# ETC: Towards Training-Efficient Video Synthesis With Exploiting Temporal Capabilities of Spatial Attention

## Abstract

Recently, synthesizing video from the text, i.e, Text-to-Video (T2V), has demonstrated remarkable progress by transferring the pre-trained Text-to-Image (T2I) diffusion models to the video domain, whose core is to add new temporal layers for capturing temporal information. However, these additional layers inevitably incur extra computational overhead, as they need to be trained from scratch on large-scale video datasets. Instead of retraining these costly layers, we conjecture whether temporal information can be learned from the original T2I model with only Spatial Attention. To this end, our theoretical and experimental explorations reveal that Spatial Attention has a strong potential for temporal modeling and greatly promotes training efficiency. Inspired by it, we propose *ETC*, a new T2V framework that achieves high fidelity and high efficiency in terms of training and inference. Specifically, to adapt the video to the spatial attention of T2I, we first design a novel temporal-to-spatial transfer strategy to organize entire video frames into a spatial grid. Then, we devise a simple yet effective Spatial-Temporal Mixed Embedding, to distinguish the inter-frame and intra-frame features. Benefiting from the above strategy that actually reduces the model's dependence on the text-video pairing dataset, we present a data-efficient strategy, Triple-Data (caption-image, label-image, and caption-video pairs) fusion that can achieve better performance with a small amount of video data for training. Extensive experiments show the superiority of our method over the four strong SOTA methods in terms of quality and efficiency, particularly improving **FVD by 49%** on average with only **1% training dataset**.

## 1 Introduction

> *"Entities should not be multiplied unnecessarily"* — *William of Ockham (1323)*

Text-to-Video (T2V) synthesis (Hong et al., 2022; Blattmann et al., 2023), generating coherent, high-fidelity video based on textual conditions, has gained great attention with a wide range of applications, such as film production and video editing. Unlike Text-to-Image (T2I) (Ding et al., 2022; Saharia et al., 2022) which only deals with static spatial information, T2V tends to be more challenging since it involves consecutive spatial representations and maintains complex temporal consistency.

Benefiting from the great breakthroughs in T2I diffusion model, recent mainstream T2V methods are to transfer the training knowledge of T2I diffusion to the video domain (Zhou et al., 2022; Chen et al., 2023; 2024). In practice, these methods, e.g., LVDM (He et al., 2022),

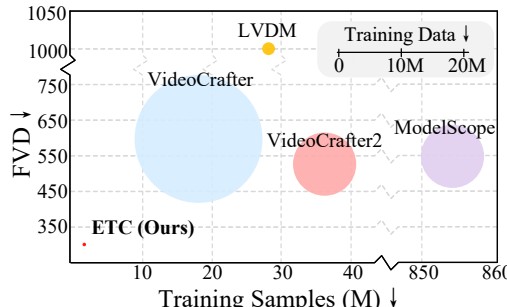

Figure 1: Comparison of FVD, training samples (= training steps × batch size), and training data of T2V diffusion models using MSR-VTT. ETC only requires **1% training datasets and 4% training samples compared to the optimal value of each metric**, with better video generation quality.

add new temporal attention layers for modeling temporal information while maintaining the original structure of the T2I diffusion model, including pre-trained parameters. However, these additional attention layers must be learned from scratch on large-scale video datasets to perform well, which inevitably brings a huge training overhead. We note that zero-shot video generation (Hong et al., 2023a; Su et al., 2023), as a special video generation task, maintains inter-frame consistency without any additional temporal module, which mainly comprises three components, including 2D Convolution, Spatial Attention, and Cross Attention. Here, the 2D convolution is independent between frames, and cross-attention is used to inject textual information. Thus, we conjecture that temporal information can be learned from the spatial attention of original T2I models.

With this conjecture in mind, we verify it in both theoretical and experimental aspects. Specifically, from the theoretical aspect, we mathematically prove that despite multiple dimensional transitions, the mapping of spatial and temporal attention remains linear without complex derivatives or power relationships, indicating that only spatial attention can model temporal information. Details are shown in Section 3. From the experimental aspect, we conduct an experiment to train a T2V model that first simply organizes the whole video in a spatial grid and then directly fine-tunes T2I models. Experimental results show that the model can produce relatively high-quality videos with only 500 steps in fine-tuning and converges at 15k steps, shown as Figure 2. Based on the above observations, we can draw an insight: spatial attention itself has a strong potential for temporal modeling, which can greatly facilitate the efficiency of model training.

Inspired by the above insight, we propose a new text-to-video synthesis model, called **ETC**, which greatly boosts high-fidelity and training efficiency. Specifically, we design a novel temporal-to-spatial transfer strategy that flattens the multi-frames into a single dimension within the spatial attention to capture temporal information. To ensure the model accurately recognizes relationships between tokens within and across frames, we introduce a simple yet effective Spatial-Temporal Mixed Embedding to distinguish between frames. With dimension changes in the latent size of noise, this embedding could support generation at any resolution or frame rate. Additionally, due to the above method, we keep the original pre-training T2I model's parameters without additional modules, thereby reducing the requirements of text-video pair datasets. To this end, we propose a Triple-Data (caption-image, caption-video, and label-image pairs) Fusion, a data-efficient strategy, to train ETC by selecting a minimal high-quality dataset. Figure 1 conducts an experiment by comparing our ETC and the other four strong baselines on the MSR-VTT dataset from three perspectives, including FVD, training samples, and training data. From this figure, we can see that our method ETC significantly improves FVD by 49%, reduces training datasets by 99%, and reduces training samples by 96%, demonstrating that the effect of our method for high-fidelity and training efficiency.

Our contributions in this paper can be summarized as follows:

- We make the theoretical and experimental exploration, which reveals that spatial attention in T2I has a strong capability of temporal modeling and can greatly boost the efficiency of training.
- We propose ETC, a novel training-efficient framework, which can produce high-quality video and avoid huge training costs.
- Extensive experiments on three datasets with zero-shot testing prove the superiority of ETC in terms of quality and efficiency.

## 2 RELATED WORK

### 2.1 TEXT-TO-VIDEO VIDEO GENERATION

In computer storage, a video is composed of multiple frames of images, and in the field of generation, video information is considered to consist of spatial information within images and temporal information between images. A common approach among video generation researchers is to build upon previously pre-trained image generation models, extending them with temporal models for video generation.

For instance, CogVideo (Hong et al., 2023b) enhances the large-scale T2I transformer CogView2 (Ding et al., 2022) by incorporating temporal information through inter-frame attention

mechanisms. In contrast, Make-A-Video (Singer et al., 2023) diverges from the typical reliance on text-video pairs for T2V generation by leveraging a pretrained T2I model, thereby eliminating the need for text-video paired training. Imagen Video (Ho et al., 2022) builds upon Imagen (Saharia et al., 2022) by employing a cascaded diffusion model that utilizes both attention and convolution across multiple resolutions. Furthermore, as the quality of video generation improves, recent research has begun to explore various settings for generation. For example, Tune-A-Video (Wu et al., 2022) introduces a one-shot video tuning method for T2V generation, incorporating temporal attention into the Stable Diffusion framework (Rombach et al., 2022b). And Text2Video-Zero (Khachatryan et al., 2023a) enables zero-shot T2V generation without training video. These works all underscore the utilization of pre-trained image generation models to supply spatial information for video generation, which is undeniably effective. However, they each introduce a new trainable module to the original model for processing temporal information. This module requires training from scratch, making the process significantly resource-intensive. And using additional modules may result in a potential waste of resources, see Appendix A and B.

## 2.2 ZERO-SHOT VIDEO GENERATION

While video generation researchers generally agree that spatial and temporal information should be processed separately to reduce computational load and preserve the quality of the original generated image, some have begun to explore merging spatial and temporal information in zero-shot settings. For instance, VidToMe (Li et al., 2023b) extends ToMe (Bolya et al., 2022) to video generation by merging video tokens into image tokens for attention processing. Similarly, another work Li et al. (2023a) employs an expectation-maximization iteration to update a basis set for temporal modeling within spatial attention. Latent-shift (An et al., 2023) propose a parameter-free temporal shift module that can leverage the spatial U-Net as is for video generation. Text2Video-Zero Khachatryan et al. (2023b) encoding motion dynamics in the latent codes, and reprogramming each frame's self-attention using new cross-frame attention. Some works use LLM as directors to process temporal information. DirecT2V (Hong et al., 2023a) utilizes LLM directors to divide user inputs into separate prompts for each frame to generate videos. Free-Bloom (Huang et al., 2024) uses LLM directors to generate high high-fidelity frames with an annotative modification LDM. These studies focus on zero-shot video generation, which tends to produce lower-quality outputs compared to models that undergo temporal tuning. In contrast, Lee et al. (2024) takes a different approach by modeling temporal information within spatial attention. It concatenates four images into a single large image and uses an original image diffusion model to handle video through autoregressive interpolation. These models do not use additional timing modules when generating videos, and complete video generation tasks such as frame insertion and video editing. Inspired by those approaches, we have designed and implemented a model that eliminates the need for the temporal module of typical text-to-video generation. We hope this work will provide valuable insights for future advancements in this scope.

## 3 OBSERVATION

The additional temporal parameters in text-to-video models bring huge training costs and require large-scale text-video datasets. We note that recent zero-shot video generation is no need for additional temporal parameters. By introducing subtle adjustments to the noise level, the layout of the generated images can be influenced to reflect temporal changes without compromising quality. This demonstrates that a well-trained image generation model is capable of generating temporal information, which can be unlocked with appropriate methods, thereby eliminating the need for separate temporal attention. So, which structure can handle the temporal information for pure spatial image diffusion? Image diffusion has three compositions: 2D convolution, Spatial Attention, and Cross Attention. Cross Attention is responsible for integrating textual information into the generation process, while the inter-frame information handled by 2D convolution remains independent. Consequently, image diffusion, with only spatial attention layers, possesses the potential to process temporal information. Therefore, we speculate that spatial attention is the module for the original T2I model to learn temporal information.

With the above conjecture, we verify it in both theoretical and experimental aspects:

**Theoretical Observation: Spatial Attention in T2I has the potential for temporal modeling**
We investigate whether the mapping established solely by spatial attention can be equivalent to that

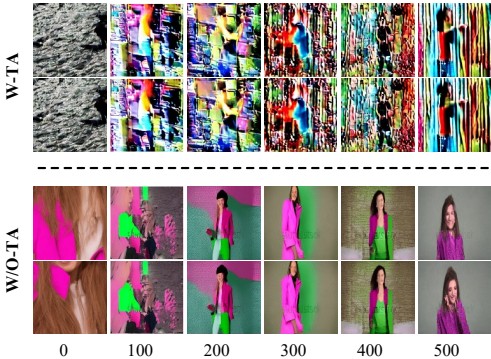 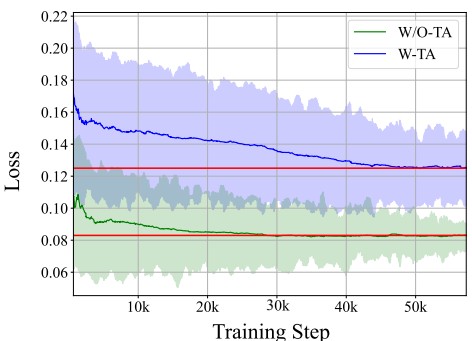

(a) Comparison of training results with and without temporal attention for steps 0 to 500. The input text is "Caucasian woman pink jacket isolated on chroma green screen background funky smiling". This indicates that without temporal attention, the model can generate videos after only 500 steps of fine-tuning.

(b) Comparsion of Loss curves with and without temporal attention. The red lines indicate the approximate convergence loss values for different models. The shaded area represents the loss range. This indicates that without temporal attention, the model converges more rapidly.

Figure 2: Key Observation: The spatial-only diffusion requires only a few finetune steps to generate video. "W-TA" and "W/O-TA" represent "with temporal attention" and "without temporal attention", respectively.

created by the combination of spatial and temporal attention. We demonstrate that spatial attention modeling a linear mapping as $\chi_s(x) = [x_1 \cdot W_s, x_2 \cdot W_s, \ldots, x_t \cdot W_s]$ and alternating between spatial and temporal attention modeling another linear mapping as $\chi_{st}(x) = \sum_{i=1}^{t} W_s^T \cdot x_i \cdot W_{Ti}$, which does not model complex derivative or quadratic relationships. Those all remain a linear combination of the input data, and therefore, a single spatial attention can be used as a substitute. (Details are shown in Section B.2)

We also find that using single spatial attention has a larger receptive field than spatial and temporal attention. When images are stitched together, the receptive field expands to encompass the entire video. (proved in B.1) In contrast, existing temporal modules limit the receptive field to small regions across different frames. Compared to the combination of spatial and temporal attention, using only spatial attention to process the entire video can theoretically increase the receptive field by a factor of the number of frames.

**Experiment Observation: The diffusion without Temporal Attention requires only a few finetune steps to generate video.** As shown in figure 2a. Diffusion With TA (W-TA) learns temporal information from scratch with new additional temporal attention, which requires more video training. And it only generates a blurry human pose even at 500 steps. In contrast, W/O-TA fine-tunes the existing module and can achieve a clear human pose in only 200 steps and produce high-quality videos in 500 steps. We also visualized the training loss curves as shown in the figure 2b. Diffusion Without Temporal Attention (W/O-TA) first reaches the convergence region around 5k and oscillates within the convergence region after 15k. In contrast, W-TA with temporal attention only reaches the convergence region after 25k. This proves that Spatial Attention can effectively utilize the pre-knowledge in Image pre-training to generate coherent videos.

In summary, since spatial attention captures temporal information during image pre-training, leveraging it for temporal modeling enhances training efficiency and effectiveness.

## 4 METHOD

The above observations indicate that spatial features possess the ability to model temporal dynamics and can be effectively utilized for video generation. Therefore, in this section, we build our spatial-only diffusion model, ETC, in 4.1. Then, to train ETC, we propose a training method with Triple-Data Fusion in 4.2.

### 4.1 THE MODIFICATION OF SPATIAL ATTENTION

To exploit the temporal capability of spatial attention, we propose a temporal-to-spatial arrangement method to enable spatial attention to process the whole video. Specifically, we stitch the video frames in the spatial dimension to train a text-to-video generation model by original image diffusion. However, this naive approach may cause a single spatial module to fail to correctly distinguish frame boundaries. Therefore, we propose Spatial-Temporal Mixed Embedding to distinguish features with inter-frame and intra-frames.

#### 4.1.1 TEMPORAL-TO-SPATIAL TRANSFER

To process videos using spatial modules, we concatenate multiple video frames along the spatial dimension. Considering a video $V \in \mathbb{R}^{B,T,h,w,C}$, should be processed by ETC. We unfold the video along the $T$ dimension, distributing $\sqrt{T}$ video frames across the $h, w$ dimensions. These frames are then concatenated into a single image $I \in \mathbb{R}^{B,H,W,C}$, where $H = h \times \sqrt{T}$ and $W = w \times \sqrt{T}$, with the position of each video frame within the image $I$ defined by the following formula:

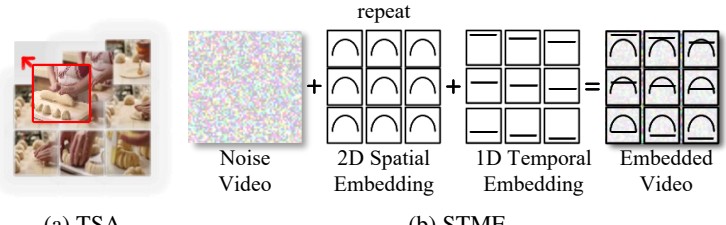

(a) TSA        (b) STME

Figure 3: The schematic diagram of how to modify spatial attention. (a) Temporal-to-Spatial Arrangement. (b) Spatial-Temporal Mixed Embedding.

$$I_{(x,y,c)} = V^{\lfloor \frac{x}{h} \rfloor + \lfloor \frac{y}{w} \rfloor \times \sqrt{T}}_{(x \bmod h, y \bmod w, c)} \tag{1}$$

where $I_{x,y,c}$ represents the point at coordinates $(x,y)$ in the image $I$, while $V^t_{(x,y,c)}$ represents the point at coordinates $(x,y)$ in the $t$-th frame of the video $V$.

#### 4.1.2 SPATIAL-TEMPORAL MIXED EMBEDDING (ME)

When the spatial and temporal information of a video is compressed into a single dimension, the network lacks modules other than convolutional layers that can differentiate between spatial and temporal aspects. As a result, a single spatial module may fail to accurately discern the boundaries between different frames, potentially leading to incorrect images. Additionally, to support multi-resolution and multi-frame rate generation, we designed the Spatial-Temporal Mixed Embedding (ME). This module consists of two parts: a 2D spatial position embedding $ME^{Sp}$ and a 1D temporal position embedding $ME^{Te}$. Both embeddings are constructed using a combination of sine and cosine functions. The module is defined as follows:

$$ME^{Sp}_{(x,y,c)} = sin(\frac{x}{\Theta^{\frac{c}{C}}}) + cos(\frac{y}{\Theta^{\frac{c}{C}}}), \tag{2}$$

$$ME^{Te}_{(x,y,c)} = sin(\frac{(x \times X) + y}{\Theta^{\frac{c}{C}}}) + cos(\frac{(x \times X) + y}{\Theta^{\frac{c}{C}}}) \tag{3}$$

where $x$, $y$, and $c$ stand the 2 image dimensions and channel dimension of the current encoded image patch, and $X$, $Y$, and $C$ stand the total of them.

Directly adding the spatial (sp) and temporal (te) embeddings can result in tokens at different positions having the same position embedding, as proven in the appendix. To prevent different tokens from sharing identical position embeddings, we add the sp and te embeddings to different noise dimensions, ensuring their independence and eliminating this overlap. The combination method is as follows:

$$I^{Sp\text{-}Te}_{\text{video}(x,y,c)} = \chi(c < \lfloor C/2 \rfloor) \cdot I^{Sp}_{\text{video}(x,y,c)} + \chi(c \geq \lfloor C/2 \rfloor) \cdot I^{Te}_{\text{video}(x,y,c-\lfloor C/2 \rfloor)} \tag{4}$$

where $\lfloor x \rfloor$ represents the greatest integer number smaller than $x$. $\chi(A)$ represents a Boolean condition function. When $A$ is true, $\chi(A)$ equals 1; otherwise, $\chi(A)$ is 0. For detailed proof, please refer to the Appendix Section C.3.

To generate videos with different resolutions or frames, we only need to adjust the dimensions of the input noise according to the image stitching method described above. The ME can then automatically adapt to the noise of varying frame rates and image sizes, producing the target video. Any changes in resolution require a warmup process of several hundred steps. Additionally, the ME must be added to the noise before any attention or convolutional modules.

## 4.2 TRIPLE-DATA DRIVEN TRAINING (TDT)

Because we eliminate the additional temporal attention, the data required for ETC is greatly reduced. We propose Triple-Data Fusion to train ETC with image-video mixed training data using a selected high-quality video dataset.

**FPS Embedding.** To enhance the spatial module's understanding of temporal information in videos and to support multi-frame rate video and image-video mixed training, we propose an FPS Embedding module. Given that the maximum timestep in diffusion models is 1000, which is sufficient to model the highest frame rate in the dataset, we share parameters between the FPS embedding module and the timestep embedding module. Before each timestep begins, the FPS value and timestep are processed through the same embedding

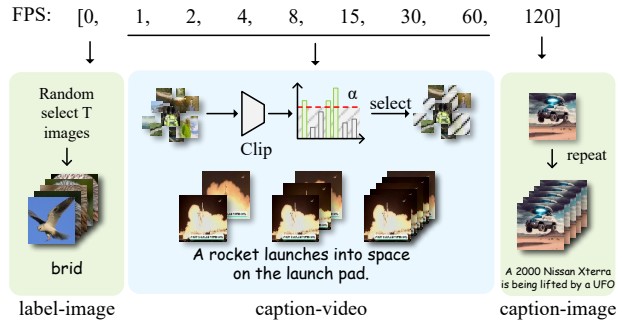

Figure 4: The schematic diagram of Triple-Data Driven Training.

module, after which they pass through distinct learnable linear modules to obtain their respective embeddings. These embeddings are then directly added to the noise input of the U-Net. Specifically, the FPS range is restricted to 0 to 120, with the FPS value for each training instance randomly selected and the FPS value for inference predetermined. The linear module consists of three linear layers, where the channel dimension is first increased fourfold and then reduced back to the original dimension.

**Video Filter.** Since our model requires only a minimal amount of data for training, it is crucial to filter high-quality videos from low-quality video datasets. ETC employs CLIP as the text feature extractor, meaning that the CLIP similarity can partially influence the generation model's capability to understand the video with the multimodal context. Low-quality videos generally fall into two categories: 1) some scenes in a video that do not align with the text description, and 2) videos with meaningless or unclear frames. The first issue can be reflected by the similarity between different modality features in CLIP, while the second issue influences CLIP's ability to extract image modality features, and results in a bad CLIP score. Therefore, we compute the CLIP score to measure the similarity between the video and its caption on a frame-by-frame basis, which can be expressed as follows:

$$CLIP_{Score} = \varepsilon \cdot \frac{CLIP_{img}(\text{Image})}{\|CLIP_{img}(\text{Image})\|_2} \cdot \left(\frac{CLIP_{text}(\text{Caption})}{\|CLIP_{text}(\text{Caption})\|_2}\right)^T \tag{5}$$

where $\|x\|_2$ denotes the L2 norm, defined as $\sqrt{\sum_i x_i^2}$. $CLIP_{img}$ and $CLIP_{text}$ represent the image and text feature extractors in CLIP, respectively. $\varepsilon$ is a constant, equal to $\ln(1/0.07)^e$. $x^T$ represents the matrix transpose of $x$.

After calculating the CLIP score for all videos, the dataset selection process involves selecting a threshold ratio $\alpha$ and excluding all videos with scores below this threshold. By adjusting the $\alpha$ value, datasets with varying quantities of high-quality videos can be obtained.

**Triple-Data Training.** To enable the model to support training on image datasets, we propose the Triple-Data Training strategy.

With the FPS Embedding and Video Filter equipped. We next train ETC with triple-data, which are label-image data, video data, and caption-image data. Since the highest frame rate in the dataset we use is 60 FPS, we set the FPS values for video training between 1 and 60. For convenience in

| Model | Data ↓ | Param ↓ | Speed ↑ | Training Samples | | MSR-VTT | | UCF-101 | VC | | |
| | | | | iter ↓ | batch ↓ | FVD ↓ | CLIP ↑ | FVD ↓ | CLIP ↑ | User Study ↑ |
|---|---|---|---|---|---|---|---|---|---|---|
| LVDM | 2M | 1.0B | 0.63 | 432K* | 64* | 999 | 29.19 | 985 | 28.44 | 2.14% |
| VideoCrafter | 20M | 1.2B | 0.43 | 136K | 128 | 567 | 27.59 | 881 | 29.48 | 8.40% |
| VideoCrafter2 | 10M | 1.4B | 0.47 | 270K | 128 | 527 | 28.71 | 700 | 29.88 | 34.69% |
| ModelScope | 10M | 1.3B | 0.55 | 267K | 3200 | 550 | 29.30 | 660 | 30.01 | 19.33% |
| ETC (Ours) | **0.1M** | **0.9B** | **1.92** | **15K** | **48** | **326** | **31.01** | **612** | **30.49** | **35.44%** |

Table 1: Qualitative comparisons with four strong SOTA. Because LVDM does not indicate the training details, "*" is an estimated value.

frame extraction from the dataset, we use only 7 discrete FPS values for training, which are [1, 2, 4, 8, 15, 30, 60]. Additionally, we use two different types of image datasets: 1) label-image datasets, where a label may correspond to multiple images; and 2) caption-image datasets, where a caption corresponds to a single image. For label-image datasets, we select a label and randomly choose a number of images corresponding to that label to create training videos. During training, we set the FPS to 0. An FPS of 0 means that the time difference between frames is infinitely large, so only the spatial correctness between frames needs to be preserved, without temporal coherence. Conversely, for caption-image datasets, we repeat the image multiple times to create a completely identical video and set the FPS to 120. The 120 FPS is the maximum frame rate in our model, meaning that the differences between images in such a short time are negligible and can be ignored. This process can be described as follows:

$$V^t = \begin{cases} I^t & FPS = 0 \\ V^{\frac{60}{t} \times FPS} & 1 \le FPS \le 60 \\ I^0 & FPS = 120 \end{cases} \quad (6)$$

where $I^t$ denotes the $t$-th image of certain label or text.

## 5 EXPERIMENT

### 5.1 SETTINGS

**Datasets.** Follow the settings of previous text-to-video generation models, we select two close-domain public video datasets and create an open-domain dataset for testing, including a) **MSR-VTT** (Xu et al., 2016), a caption-video pair dataset, b) **UCF-101**, an action recognition dataset, which contain label-to-video pairs, and c) **VC** Video Caption dataset with 500 prompts, which consists of full sentences generated by ChatGPT (OpenAI, 2021).

**Metrics.** To comprehensively evaluate the effectiveness and efficacy of different text-to-video generation models, we adopt four commonly used metrics as follows: a) **FVD** (Unterthiner et al., 2018), which is pertained by Kinetics (Kay et al., 2017) dataset to evaluate the quality of spatial and temporal features in video generation, b) **Clip-Score** Hessel et al. (2021), which is to measure the alignment of text and image denoted as CLIP, c) **User Study** to measure the human-like, and d) **Speed**, which aimsto provide an assessment of the practical running speed by frame per second.

**Training Details.** The spatial modules are initialized with weights of SD2.1 (Rombach et al., 2022a). The base training resolution is set to $256 \times 256$ at 16 frames. We utilize the selected WebVid-0.1M (Bain et al., 2021), ImageNet (Deng et al., 2009) and JDB (Sun et al., 2024) datasets. This model is trained on 8 NVIDIA 3090 GPUs for 15K iterations with a batch size of 48. The learning rate is set to $1 \times 10^{-4}$ for all training tasks.

### 5.2 MAIN RESULTS

To verify the efficacy and effectiveness of ETC, we conduct comparative experiments on MSR-VTT, UCF-101, and VC datasets in zero-shot settings using the baselines including LVDM (He et al., 2022), VideoCrafter (Chen et al., 2023), VideoCrafter2 (Chen et al., 2024), and ModelScope (Wang et al., 2023b). We equally sample 10k 256x256 videos for each baseline. The quantitative results are shown in Table 1.

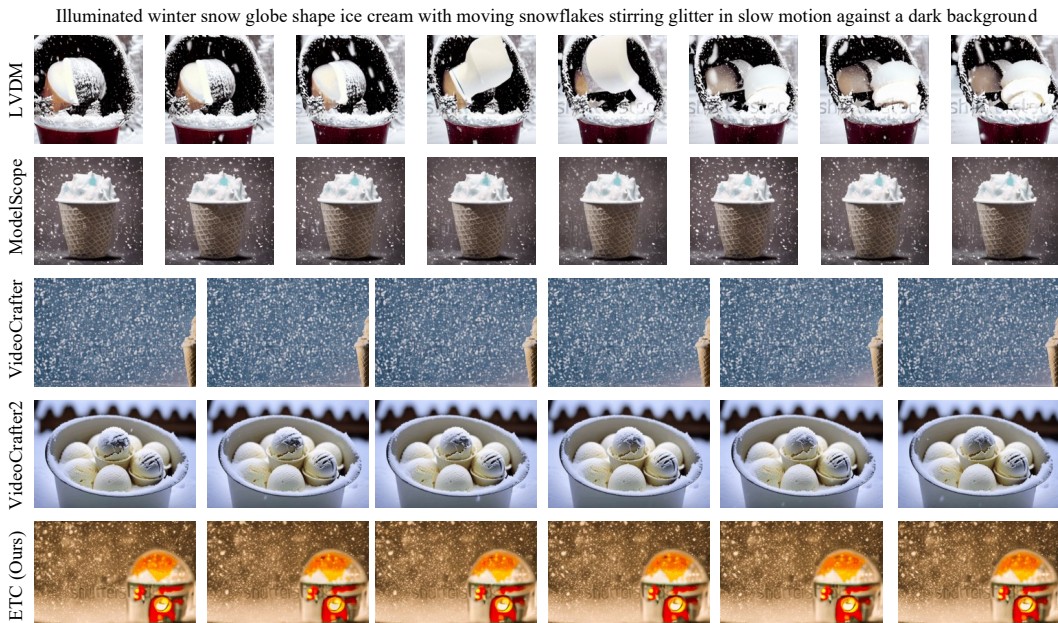

Illuminated winter snow globe shape ice cream with moving snowflakes stirring glitter in slow motion against a dark background

Figure 5: Quantitative comparisons with four baselines.

**Qualitative Results.** As indicated in the table 1, we demonstrate the effectiveness of ETC below: (1) ETC without the newly added module, which significantly reduces dataset dependency by about 99%. (2) Without the temporal module, ETC only has 0.9B parameters, which significantly acceler­ates the inference process. (3) ETC achieves a speed of 1.92 FPS, approximately three times faster than LVDM. (4) With only the spatial module, fine-tuning can be completed in just 15k steps with a batch size of 48. This significantly reduces the demand for GPUs, decreasing the training time from several GPU years to just a few GPU months. (5) There is a significant improvement in the FVD metric on two public datasets, MSR-VTT and UCF-101, indicating that the features of our generated videos are much closer to the feature distribution of these datasets. (6) The CLIP metric also shows that the correlation between our generated videos and the corresponding text is higher on the MSR-VTT public dataset. We conduct experiments on an open-domain dataset VC and prove that ETC achieves the best CLIP metric among the four baselines. (7) For the user study, we pre­sented each volunteer with five video models generated from the same text and asked them to select the best one. The table records the percentage of times each model was chosen as the best by the volunteers. The results show that ETC and VideoCrafter2 performed similarly, with VideoCrafter2 slightly outperforming ETC. The remaining three video generation models received lower scores. This indicates that, in terms of meeting human preferences, ETC can achieve comparable results to SOTA models. Finally, the user study scores also show that our results are the best among several different models, but similar to videocrafter2. We have analyzed the user study data in detail in the section F.

**Quantitative Results.** The qualitative comparison in figure 5 shows the videos generated by ETC and the other four SOTA methods. When encountering complex scenes, the generated results from LVDM, ModelScope, and VideoCrafter do not include all objects or generate low-quality video. In contrast, our ETC results can effectively include the whole scene with accurate scenes and styles. This proves that ETC has superior visual effects compared to other SOTA.

**High Resolution and Long Video Generation.** To prove the generalization ability, we conduct experiments on the scalability of ME. We finetune the well-trained ETC for each setting for 1k steps. The results are shown in the figure 6. For high-resolution experiments, we set the resolution to $512 \times 320$ with 16 frames. In order to make the images clearer, we used macro fruit slices as prompt inputs. We can see that every kiwi seed is clearly visible, and the water splashes are also sharp. For long video generation, we used a time-lapse photography prompt to generate a 256-frame video with a resolution of $256 \times 160$. The mountains remain consistent throughout, and the clouds

Sliced kiwi falling and splashing in slow motion

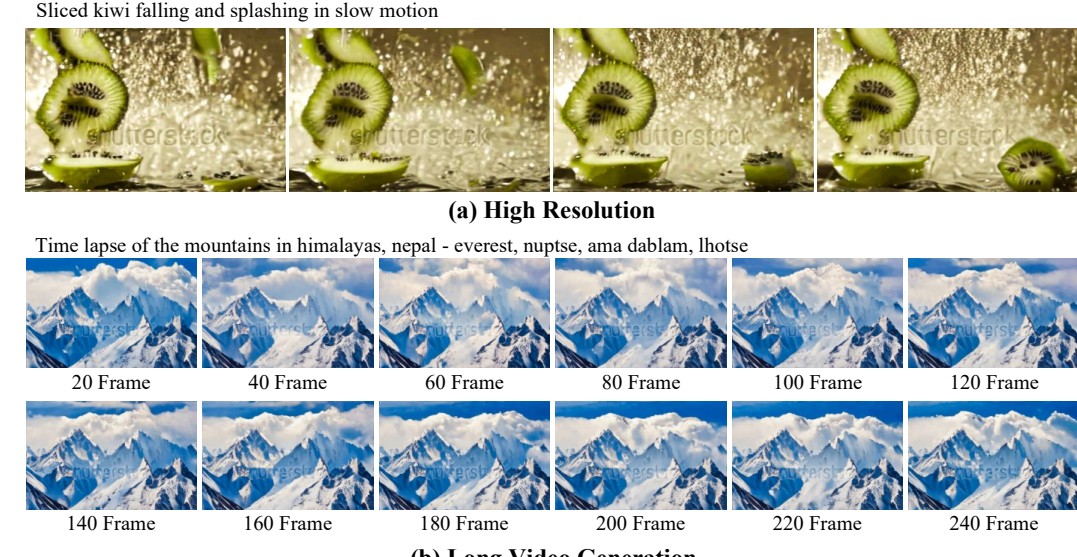

**(a) High Resolution**

Time lapse of the mountains in himalayas, nepal - everest, nuptse, ama dablam, lhotse

| 20 Frame | 40 Frame | 60 Frame | 80 Frame | 100 Frame | 120 Frame |
|---|---|---|---|---|---|

| 140 Frame | 160 Frame | 180 Frame | 200 Frame | 220 Frame | 240 Frame |
|---|---|---|---|---|---|

**(b) Long Video Generation**

Figure 6: Generalization study of ETC for (a) high resolution and (b) long video generation task in WebVid dataset.

change shape over time. This demonstrates the exceptional generalization ability of Sp-Te Position Embedding, allowing for the generation of any resolution and frame count with minimal fine-tuning.

## 5.3 ABLATION STUDY

**Ablation Study on Dataset Size.** To determine the optimal dataset size, we train the ETC with datasets of varying sizes and perform an FVD test every 3K steps, as shown in figure 7. As training progresses, the FVD of the model gradually decreases. For the dataset from 10K to 50K to 100K, it can be observed that the FVD decreases significantly. This indicates that these two dataset sizes are insufficient for training the ETC. From 100K to 300K, the trend of FVD remains almost the same. Therefore, 100K appears to be the approximate amount of data required for training the ETC, and further increases in dataset size do not significantly en-

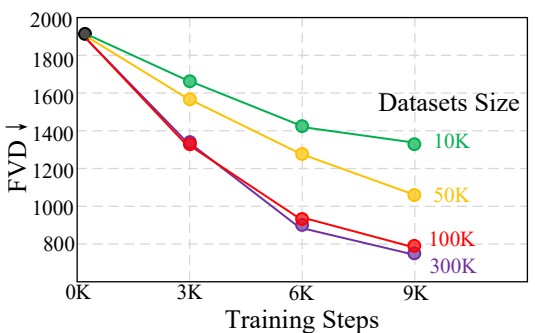

Figure 7: Ablation study on dataset size.

hance quality. Consequently, we chose a dataset size of 100K for all experiments. Additionally, all data in this ablation experiment were obtained by filtering WebVid.

**Ablation Study on ETC Modules.** To validate the effectiveness of the Spatial-Temporal Mixed Embedding (ME) and Triple-Data Driven Training (TDT) modules in our approach, we conduct ablation experiments on these two modules using the MSR-VTT dataset. As shown in table 2. We find that incorporating TDT, along with additional image datasets, improves both video generation quality and image-text alignment. The addition of ME significantly enhances video quality. Moreover, without ME, the model may incorrectly segment images in the stitched

| ME | TDT | FVD ↓ | CLIP ↑ | Err. ↓ |
|---|---|---|---|---|
| | | 462 | 27.84 | 17.4% |
| ✓ | | 396 | 30.27 | 17.8% |
| | ✓ | 377 | 28.55 | 0% |
| ✓ | ✓ | **326** | **31.01** | **0%** |

Table 2: Ablation study for ME and TDT on MSR-VTT dataset.

video (marked as Err. in table), with an error rate of approximately 17%. After adding ME, this error rate drops to 0%. Finally, we conduct experiments with both TDT and ME combined, demonstrating that the integration of these two modules achieves optimal video quality and image-text consistency.

## 6 CONCLUSION

We aim to improve video generation models by removing temporal attention and transferring its function to spatial attention. To support this approach, we propose the Spatial-Temporal Mixed Embedding, allowing the same attention mechanism to distinguish between intra-frame and inter-frame information. Additionally, we introduce the FPS-based Triple-Data Driven Training. As a result, we develop a high-quality, high-speed video generation model with minimal data dependency. We believe that our work corrects mistakes in the design of previous video generation models and will inspire future advancements in video generation.

However, unlike autoregressive models, we do not support changes in resolution and frame rate without additional training. We expect that with training on mixed resolutions, the Spatial-Temporal Mixed Embedding can enable the model to generate videos with different resolutions and frame rates. This issue will be explored in more detail in future research.

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

## A  PERFORMERS

In this chapter, we use the abstracted and simplified image and video diffusion model to comprehensively analyze the performance. First, in Section A.1, we analyze the differences in computational complexity and parameter count between image diffusion, which uses only spatial attention, and video diffusion, which incorporates both spatial and temporal attention, from a theoretical perspective. Next, in Section A.2, we further elucidate the differences in actual running time through experiments.

### A.1  THEORETICAL ANALYSIS

In this section, we conduct a theoretical analysis comparing the computational complexity and parameter count of image diffusion and video diffusion.

### A.1.1  MATHEMATICAL DEFINE

We assume that a video $V \in \mathbb{R}^{B,T,H,W,C}$ need to be generated by those two generation model. In image diffusion, we concat each video frame to a whole image $I \in \mathbb{R}^{B,H \times \sqrt{T}, W \times \sqrt{T}, C}$ using the method we proposed in Section 4. The primary computational load lies in the attention mechanisms of both image and video diffusion. To simplify the comparison, we focus only on the computation and parameter count of the spatial and temporal attention mechanisms in these two models. So we assume that only spatial attention in image diffusion and both

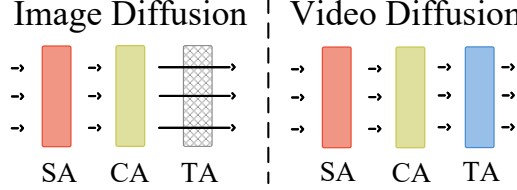

Figure 8: A schematic diagram of image diffusion (left) and video diffusion (right). where "SA" denotes spatial attention, "CA" denotes cross attention, "TA" denotes temporal attention.

spatial attention and temporal attention are in video diffusion. The schematic diagram of image and video diffusion is shown in Figure 8.

### A.1.2  ATTENTION

Both spatial attention and temporal attention are use self-attention mechanisms in the diffusion attention block. We assume that input hidden states $hs \in \mathbb{R}^{B,L,C}$ will be processed by self-attention. This process can be described as follows:

$$Q = hs \times W_Q, \quad K = hs \times W_K, \quad V = hs \times W_V, \tag{7}$$

$$hs_{out} = \text{Softmax}\left(\frac{Q \times K^T}{\sqrt{h}}\right) \times V \tag{8}$$

where $W_Q, W_K$, and $W_V$ are the parameters of this attention mechanism; $h$ is the dimension of the $Q, K$, and $V$. In attention, $W_Q, W_K$, and $W_V$ are all composed of matrices $[C, C]$. So the parameter count of attention is $3C^2$.

| Settings | Model | Spatial Attention | | Temporal Attention | | Total time |
|---|---|---|---|---|---|---|
| | | shape | Time (s) | shape | Time (s) | |
| Base (F = 16, Res = 256) | Image Diffusion | [1, 16384, 4] | 6.453 | - | - | 6.453 |
| | Video Diffusion | [16, 1024, 4] | 2.166 | [1024, 16, 4] | 0.947 | 3.113 |
| Long Video (F = 32, Res = 256) | Image Diffusion | [1, 32768, 4] | 24.710 | - | - | 24.71 |
| | Video Diffusion | [32, 1024, 4] | 4.209 | [1024, 32, 4] | 0.941 | 5.150 |
| High Res (F = 16, Res = 512) | Image Diffusion | [1, 32768, 4] | 24.710 | - | - | 24.71 |
| | Video Diffusion | [16, 2048, 4] | 9.196 | [2048, 16, 4] | 0.971 | 10.167 |
| High Quality (F = 32, Res = 512) | Image Diffusion | [1, 65536, 4] | 99.212 | - | - | 99.212 |
| | Video Diffusion | [32, 2048, 4] | 18.252 | [2048, 32, 4] | 1.571 | 19.823 |

Table 3: Comparison of 1,000 times Inference in four Different Settings: Image Diffusion vs. Video Diffusion.

For calculation, the steps can be described as follows:

a) $Q$, $K$, and $V$ calculation: Each of those three matrices calculation, input and output shape is $[B, L, C] \times [C, C] \rightarrow [B, L, C]$. So the calculation amount of this step is $3 \times 2BLC^2 = 6BLC^2$.

b) $Q \times K^T$ calculation: The input and output shape is $[B, L, C] \times [B, C, L] \rightarrow [B, L, L]$, which is called attention score. This matrix means how the tokens of $K$ attention to $Q$. So the calculation amount of this step is $2BL^2C$.

c) Score $\times V$ calculation: The input and output shape is $[B, L, L] \times [B, L, C] \rightarrow [B, L, C]$. So the calculation amount of this step is $2BLC^2$.

Overall, the calculation amount of whole self attention is $6BLC^2 + 2BL^2C + 2BLC^2 = 2BLC(4C + L)$.

### A.1.3 COMPARISON OF IMAGE AND VIDEO DIFFUSION

We simplify the typical image diffusion model, Stable Diffusion (Rombach et al., 2022b), which has only spatial attention, for image diffusion analysis. We assume that a whole image $I \in \mathbb{R}^{B, H \times \sqrt{T}, W \times \sqrt{T}, C}$ should be generated in this image diffusion. The image is reshaped to $[B, T \times H \times W, C]$, and the calculation amount is $2BTHWC(4C + THW)$.

In the typical video diffusion model LVDM He et al. (2022), which contains both spatial and temporal attention, the spatial attention input shape is $[B \times T, H \times W, C]$, so the calculation amount of it is $2BTHWC(4C + HW)$. The temporal attention input shape is $[B \times H \times W, T, C]$, so the calculation amount of it is $2BTHWC(4C + T)$. Therefore, the whole calculation amount of video diffusion is $2BTHWC(8C + HW + T)$.

For parameter count, image diffusion has $6C^2$ parameters due to one attention mechanism, while video diffusion has $12C^2$ parameters due to two attention mechanisms.

Overall, image diffusion has $\frac{4C+THW}{8C+HW+T}$ times the calculation amount of video diffusion. In our typical settings where $C = 4$, $(H, W) = (128, 128)$, and $T = 16$ (representing a video with 16 frames at $256 \times 256$ resolution), image generation requires about 16 times the calculation amount of video generation. For parameter count, video diffusion requires twice as many parameters as image generation.

### A.2 EXPERIMENT ANALYSIS

As mentioned in the previous section, although image diffusion has only half the number of parameters compared to video diffusion, its computational effort is 16 times greater under basic settings. To better compare their actual runtime efficiency, we conducted detailed experiments to measure the actual inference time of spatial and temporal attention in both models. The experimental results are presented in the Table 3. We tested video generation with frame counts ranging from 16 to 32 and resolutions from 256 to 512 in four different settings. In the base set, the shape is converted from the original video diffusion $[16, 1024, 4]$ to image diffusion $[1, 16384, 4]$. Although the $B \times L$ remains constant, the increase in $L$ leads to a longer inference time. The time consumed by temporal

attention also supports this conclusion. When $L$ is reduced to only 16, the inference time decreases to 0.947 seconds. Despite this, the total runtime only increases by a factor of two. Other settings involving longer videos and higher resolutions also support this conclusion. While the computational effort for image diffusion is 16 times that of video diffusion in basic settings, it only takes twice as long.

During the actual diffusion process, because each attention block in the U-Net is in the convolutional layers, the spatial dimensions $H$ and $W$ of the attention block closer to the middle of the U-Net are smaller. As a result, the computational and runtime differences between image diffusion and video diffusion are smaller. In practical runs, image diffusion with the same number of steps will be much faster than video diffusion. Referring to the main text table, typical image diffusion with Stable Diffusion is nearly 3 times faster than typical video diffusion LVDM (see Section 5.2).

# B ATTENTION ANALYSIS

## B.1 RECEPTIVE FIELD

Each attention mechanism has a receptive field for the data it processes. Suppose we have data of shape $[B, L, C]$ that needs to be processed by the attention mechanism described in Equation 8. Without the attention map, the maximum receptive field of each token is $L$, and tokens in different $B$ dimensions are independent of one another. In video processing, the original video diffusion model utilizes two attention mechanisms: (1) spatial attention to process spatial information within an image, and (2) temporal attention to process temporal information across images. The schematic diagram of those two types of attention is shown in Figure 9.

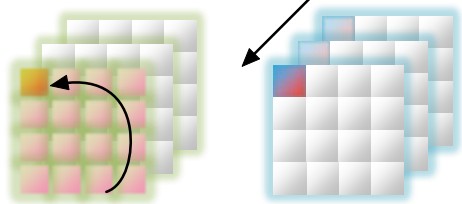

Spatial Attention    Temporal Attention

Figure 9: The schematic diagram of the receptive field of spatial attention (left) and temporal attention (right). Dark colors represent current tokens, light colors represent perceived tokens, and gray colors represent independent tokens.

The spatial attention mechanism takes an input of size $[B \times T, H \times W, C]$ for a video with $T$ frames. For each token within an image, the receptive field is limited to itself. In this spatial attention setup, no token within an image can perceive tokens in other images. Moreover, the temporal attention mechanism takes an input of $[B \times H \times W, T, C]$ for "long strip" tokens across frames. These tokens originate from different frames but occupy the same spatial positions. Consequently, the receptive field of temporal attention consists of tokens at corresponding positions in different frames, while tokens at different positions within the same frame and across different frames remain independent. For video processing in image diffusion, only one attention mechanism takes an input of $[B, T \times H \times W, C]$ for a grid video within an image. The receptive field of image diffusion encompasses the entire video. Specifically, the receptive field is calculated by $\frac{L}{T \times H \times W}$. This means:

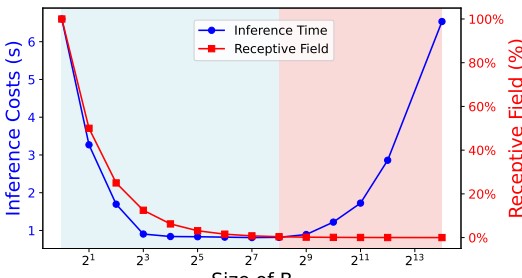

Figure 10: The compare of 1,000 times inference costs and receptive field of different $B$ in shape. The $B \times H \times W$ is a constant of $16384\ (2^{14})$. In our base setting, $B = 1\ (2^0)$ in the attention of image diffusion. $B = 16\ (2^4)$ and $B = 1024\ (2^{10})$ in the spatial and temporal attention in video diffusion, respectively. The blue shaded area indicates the part where the inference cost is lower than the receptive field, and the red shaded area indicates the opposite.

1) If a motion is too large, the spatial information between different frames may not be captured in video diffusion because of the limitation of the receptive field of temporal attention. However, any motion can be detected in image diffusion.

2) As the figure 10 shows when $T \times H \times W$ is a fixed value of 16384 ($2^{14}$), putting more tokens into the batch $B$ dimension will make the receptive field smaller and the amount of computation will first decrease and then increase.

Therefore, before the intersection of the two curves (around $1024$, $2^8$ ), the computational cost of attention is slightly smaller than the receptive field. For instance, in the range of $B = 1$ to $B = 8$ ($2^3$), the computational cost of attention is minimized, while the receptive field is still maintained at approximately 12.5%. Within this range, the computation of attention has an advantage over the receptive field. As $B$ increases, particularly beyond $512$ ($2^8$), the batch dimension becomes large, requiring the computation of numerous small matrices. Consequently, the computational cost rises sharply, while the receptive field decreases. This results in an inefficient use of computational resources for attention calculation. In our base setting, a video typically consists of around 16 frames. Therefore, the dimension of $B$ in temporal attention is approximately $1024$ ($2^{10}$) leading to a relatively long computation time for this temporal attention, about 5.4% of the maximum computation time, with an extremely small receptive field of only 0.09%. This represents a significant inefficiency in computational resource utilization.

## B.2 TEMPORAL MODELING CAPABILITY FOR SPATIAL ATTENTION

To demonstrate that spatial attention is capable of modeling temporal information, we begin by providing a mathematical proof that the traditional spatial and temporal processing approach is a linear combination of the input video, rather than a quadratic or derivative relationship, which allows it to be replaced by a single spatial mechanism in this section.

Attention (Vaswani, 2017) transforms input data into query, key, and value (QKV) representations and processes them using the Scaled Dot-Product Attention, as described by Eq. 8. This mechanism captures relationships within the same data source (Self-Attention) or across different sources (Cross-Attention). While it involves a nonlinear softmax operation, the core of the attention mechanism is largely driven by linear operations. Building on this, Schlag et al. (2021) proposed a linear Transformer model that achieves efficiency gains by linearizing the Attention mechanism. Although their primary focus was on improving computational efficiency, their work also demonstrated that Attention can, in some cases, be approximated or replaced by linear operations. Furthermore, Zheng et al. (2022) explored the linearization of self-attention mechanisms, proposing a novel method that maintains performance while reducing computational complexity to linear. Therefore, for simplification, we consider attention as a linear relationship.

Suppose we need to process a video $1 \times (t \cdot n)$ matrix $\chi = [x_1, x_2, x_3, \ldots, x_t]$ with $t$ frames, where $x_i$ represents the $i$-th frame of the video, expressed as a row vector with a length equal to the number of tokens $n$. According to the previous assumption, attention can be considered as a linear relationship, and the attention operation can be expressed as $x \cdot W$. Let the spatial linear mapping be $x \cdot W_s$, and similarly, the temporal attention operation can be expressed as $x \cdot W_t$. Where $W_s$ and $W_t$ are matrices with shapes $n \times n$ and $t \times t$, respectively. Below, we will investigate the relationship between spatial and temporal stacking in terms of information processing.

Firstly, spatial attention can be described as:

$$\chi_s = [x_1, x_2, x_3, \ldots, x_t] \times \underbrace{\begin{bmatrix} W_s & 0 & \ldots & 0 \\ 0 & W_s & \ldots & 0 \\ \vdots & \vdots & \ddots & 0 \\ 0 & 0 & \ldots & W_s \end{bmatrix}}_{t \text{ times}} = \underbrace{[x_1 \cdot W_s, x_2 \cdot W_s, \ldots, x_t \cdot W_s]}_{t \text{ times}} \quad (9)$$

Next, for the temporal operation, we need to perform attention on each corresponding module in the image $x$. Therefore, we need to transpose each image, resulting in:

$$\left[ (x_1 \cdot W_s)^T, (x_2 \cdot W_s)^T, \ldots, (x_t \cdot W_s)^T \right] = \left[ W_s^T \cdot x_1^T, W_s^T \cdot x_2^T, \ldots, W_s^T \cdot x_t^T \right] \quad (10)$$

where $x^T$ denotes the matrix transpose of $x$. Therefore the next temporal processing can be described as:

$$\chi_{st} = \left[W_s^T \cdot x_1^T, W_s^T \cdot x_2^T, \ldots, W_s^T \cdot x_t^T\right] \times W_T \tag{11}$$

$$= \left[W_s^T \cdot x_1^T, W_s^T \cdot x_2^T, \ldots, W_s^T \cdot x_t^T\right] \times \begin{bmatrix} W_{T1} \\ W_{T2} \\ \vdots \\ W_{Tn} \end{bmatrix} \text{ ($n$ times)} \tag{12}$$

$$= \sum_{i=1}^{t} W_s^T \cdot x_i \cdot W_{Ti} \tag{13}$$

where $W_{T_i}$ denotes the $i$-th row of $W_T$.

Thus, we consider two cases. If $W_s$ and $W_T$ are nonsingular matrices, then there is a linear relationship between $\chi_{st}$ and $\chi$, which can be fitted by another well-trained $W_s$. If $W_s$ and $W_T$ are singular matrices, then $\chi_{st}$ is a low-dimensional space mapping of $\chi$, and therefore, it can also be fitted by another separate $W_s$. In summary, although the dimensions of the video being processed are continually swapped between spatial and temporal processing, this does not introduce complex derivative or nonlinear relationships, which makes it possible to model this relationship using a single spatial component.

### B.3 Exploring Spatial Attention in ETC

To explore the attention mechanism of ETC, we visualize the attention distribution among frames in the mid-block of each diffusion sampling step in figure 11. The attention patterns can be broadly classified into the following categories:

- Most of the attention is on the **self frame**. This type of pattern tends to be brighter along the diagonal, with the upper and lower triangles close to black, as seen in the earlier layers in the figure.

- Most of the attention is on the **cross frames**. In this type of image, the diagonal may be bright, but the upper and lower triangles are close to green, as observed in the later layers of the figure. (This is because temporal attention is shared across multiple frames, and the average attention score allocated to each frame is relatively small. When the temporal attention of a particular frame approaches green, the total temporal attention score for that frame is already quite large.)

At lower timesteps, the frames are predominantly black, indicating that the generation process focuses more on intra-frame information. As the timesteps increase, inter-frame information starts to appear in green, suggesting that the generation model begins to focus on inter-frame information. This observation is similar to the conclusions drawn by the CogVideo (Hong et al., 2022) video generation model. This proves that although ETC only utilizes the spatial module, the attention given to spatial and temporal aspects within the spatial module at different timesteps is similar to that of a generation model with a temporal module. This further demonstrates that spatial can replace temporal in completing the generation process.

However, an interesting phenomenon can be observed in the figure: the attention of the first x heads at each timestep is concentrated on the xth frame itself. One possible explanation is that the processing of information in different heads may develop a certain degree of independence during the training process. However, the exact reason remains unclear.

## C Position Embedding

In a typical image or video diffusion model, each attention module operates within a specific processing range, as discussed in Section B.1. For instance, in the image diffusion models, the attention mechanism is confined to spatial information, meaning that the entire attention process in image diffusion is purely spatial. In the original video diffusion, researchers use off-the-shell spatial attention

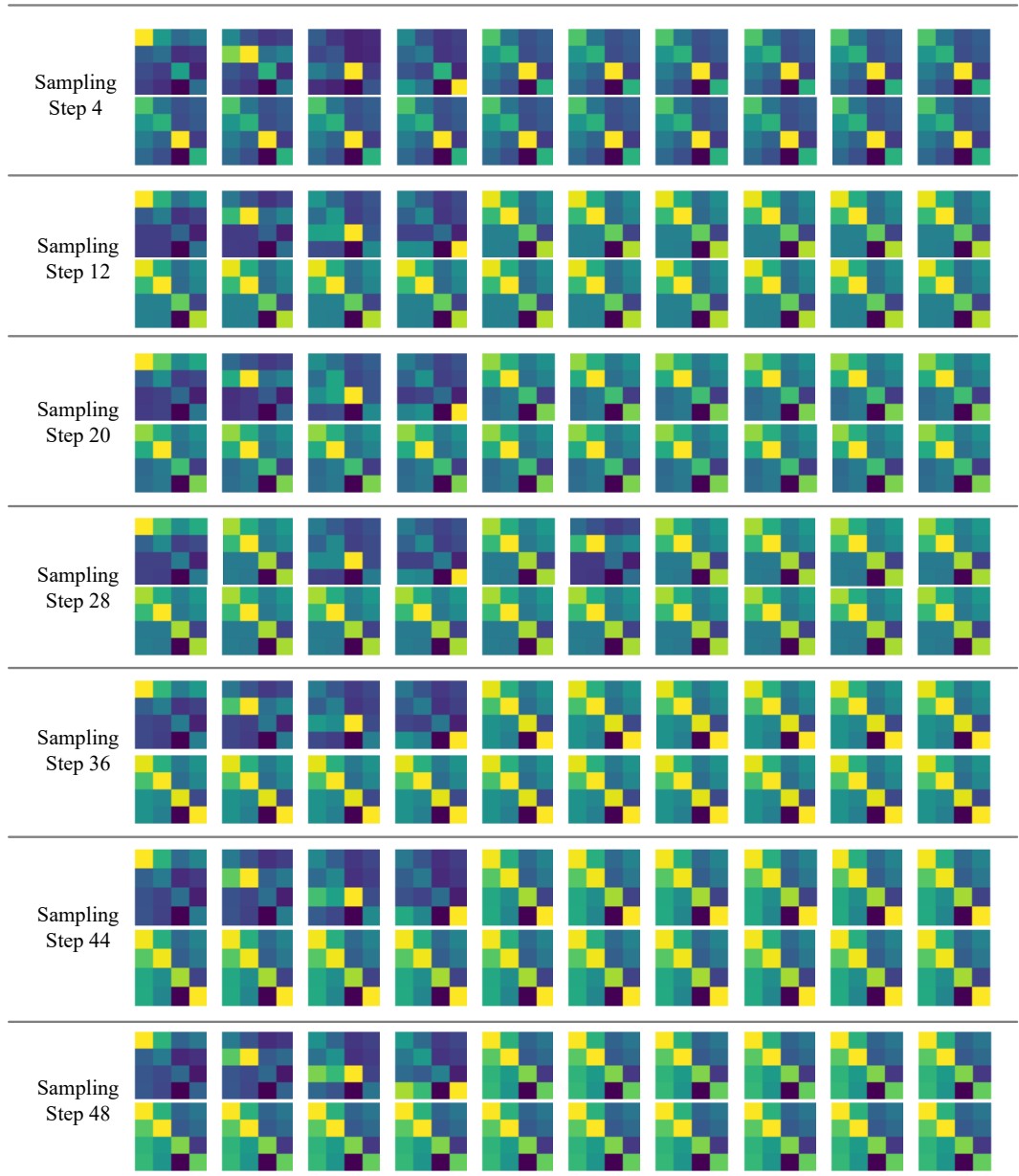

Figure 11: The attention distribution among frames in ETC generation. We use DDPM as the sampler and sample 50 steps. Only 7 steps with 20 attention heads in each step are selected for display purposes. Each attention head is visualized with a heat map of size $4 \times 4$, where a lighter yellow color represents a larger value. The $4 \times 4$ block indicates the sum of attention scores (after softmax) between each pair of frames. That is to say, the grid in row $i$ column $j$ represents $\sum_{x \in \text{Frame}_i, y \in \text{Frame}_j} \text{attn}_{x,y}$, where $\text{Frame}_i$ denotes the set of tokens in the $i$-th frame and the $\text{attn}_{x,y}$ denotes the attention score of token $x$ to $y$. In particular, we only visualize the attention distribution for mid-block.

of image diffusion and design new temporal attention to process temporal information. A permute operation has been added to the processed data, which forces control of the different receptive fields in different attention mechanisms. Different from that, we fuse the spatial and temporal into an image and process it in whole spatial attention. Although the convolutional layer has a certain de-

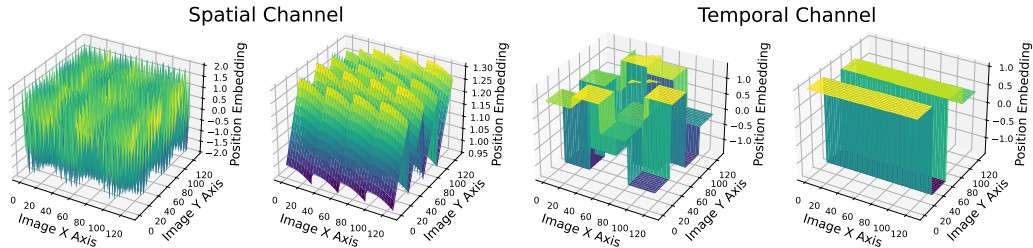

Figure 12: The visualization of our position embedding method. We use the base settings ($H = 128, W = 128$) as the input, and channel $C = 4$ for simple visualization. We add spatial position embedding (Equation 18) to the first half of channels (left) and temporal position embedding (Equation 19) to the last half of channels (right) by Equation 25.

gree of position awareness, it is still possible that the spatial attention layer cannot separate different images well. As a result, some images are incorrectly cut in our base model, see Figure 13.

To distinguish between spatial and temporal attention within the same attention module, we added position embeddings to the data. This helps guide the model in differentiating between spatial and temporal processing. Additionally, we constrained the model to perform spatial processing within specific regions to minimize confusion regarding the positions of the segmented smaller images.

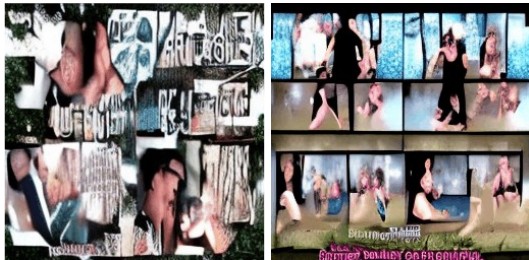

First, we define some common mathematical concepts in Section C.1. Next, we describe the typical 1D and 2D position embedding in Section C.2 and discuss the conflicts that arise from applying it twice to the image. Finally, we present our improvements to the scheme in Section C.3.

Figure 13: Some bad cases of incorrectly cut in our base model. We have cut the entire image into small images according to the frames. The cut image should be a single image. However, some generated images contain about 4 small images (left), or more irregular small images (right).

## C.1 MATHEMATICAL DEFINE

For the convenience of definition, in this chapter, we call the image formed by splicing the entire video to generate image $I_{\text{video}} \in \mathbb{R}^{H,W,C}$, and the image of each frame in the video is called image $I_{\text{frame}} \in \mathbb{R}^{h,w,C}$, where $h = H/T, w = W/T$, $T$ is the frame count of video. The location of $I_{\text{frame}}$ in $I_{\text{video}}$ can be describe as:

$$I_{\text{video}}[X, Y, C] = I_{frame}^{(Y \bmod \sqrt{T}) \times (X \bmod \sqrt{T})} \left[ X \bmod \sqrt{T}, Y \bmod \sqrt{T}, C \right] \tag{14}$$

where $mod$ stands for modulus operation ($a \bmod b = a - \lfloor \frac{a}{b} \rfloor$), $I_{frame}^i$ stands the $i$-th frame in $I_{\text{video}}$.

## C.2 FROM 1D TO 2D POSITION EMBEDDING

**1D Position Embedding.** For 1D data $d \in \mathbb{R}^{L,C}$, such as text, the relationship between data ($L_i$, $L_j$, where $i \neq j$) does not exist in 2D relation. So the isolated sinusoidal embedding of sin and cos is used in the traditional 1D data. For each position $x$ and dimension $i$, 1D data-position embedding can describe as:

$$ME_{(x,2i)}^{1D} = sin(\frac{x}{\Theta^{\frac{2i}{C}}}), \tag{15}$$

$$ME_{(x,2i+1)}^{1D} = cos(\frac{x}{\Theta^{\frac{2i}{C}}}), \tag{16}$$

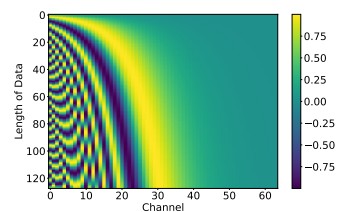

Figure 14: The visualization of 1D position embedding (Equation 15).

where $\Theta$ stands for a big number, $i$ stands index of dimension, $c$ stands the dimension of encode vector. The visualization of 1D position embedding is shown in Figure 14.

**2D Position Embedding.** In order to meet the requirements of the spatial relation of the image, we connect the sin and cos for 2D position embedding. Our basic 2D position embedding can be described as:

$$ME^{2D}_{(x,y,c)} = sin(\frac{x}{\Theta^{\frac{c}{C}}}) + cos(\frac{y}{\Theta^{\frac{c}{C}}}), \tag{17}$$

where $x$ and $y$ stands for the different dimension of that image, and $c$ stands for the channel. The visualization of 2D position embedding is shown in Figure 15.

### C.3 Spatial and Temporal Position Embedding

**Spatial and Temporal Split.** To address the issue highlighted in Section C where the model struggles to clearly distinguish image boundaries, we implement two distinct position encodings for video processing in image diffusion as Figure 16. For each individual image $I_{\text{frame}}$, we apply a two-dimensional positional encoding $ME^{2D}$ in Equation 17, which facilitates the model in learning the spatial relationships of internal features within the image. Conversely, for the entire video $I_{\text{video}}$, the frames lack a two-dimensional relationship, exhibiting only a linear sequential relationship such as ... $I^{i-2}$, $I^{i-1}$, $I^{i}$, $I^{i+1}$, $I^{i+2}$ ... Therefore, artificially stitching frames into a single image creates a pseudo-two-dimensional relationship, which does not represent a genuine prior for video processing. Our basic approach for spatial and temporal position embedding is outlined as follows:

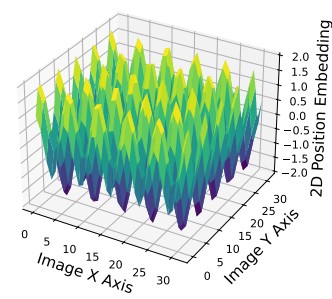

Figure 15: The visualization of 2D position embedding (Equation 17).

$$ME^{Sp}_{(x,y,c)} = sin(\frac{x}{\Theta^{\frac{c}{C}}}) + cos(\frac{y}{\Theta^{\frac{c}{C}}}), \tag{18}$$

$$ME^{Te}_{(x,y,c)} = sin(\frac{(x \times X) + y}{\Theta^{\frac{c}{C}}}) + cos(\frac{(x \times X) + y}{\Theta^{\frac{c}{C}}}) \tag{19}$$

where $x$, $y$, and $c$ stand the 2 image dimensions and channel dimension of the current encoded image patch, and $X$, $Y$, and $C$ stand the total of them.

**Spatial and Temporal Fusion.** We begin by defining the scope of spatial and temporal effects as outlined in Equation 14. For each spatial embedding, the scope is each $I_{\text{frame}}$. And for temporal embedding, the scope is global $I_{\text{video}}$. Therefore, the spatial-temporal fusion can be divided into 2 steps:

1). Adding spatial position embedding to each $I_{\text{frame}}$: According to the grid split in Equation 14, we gradually add SAME spatial position embedding (Equation 18) in each $I_{\text{frame}}$. This step can be outlined as follows:

$$I^{Sp}_{\text{video}(x,y,c)} = ME^{Sp}_{(x \bmod (X / \sqrt{T})),(y \bmod (Y / \sqrt{T})),c)} \tag{20}$$

In this way, the $T$ same spatial position embeddings are added into $I_{\text{video}}$ grid.

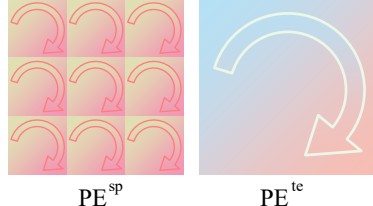

$PE^{sp}$      $PE^{te}$

Figure 16: The schematic diagram of spatial and temporal position embedding.

2). Adding temporal position embedding to global $I_{\text{video}}$: We regard each $I_{\text{frame}}$ as an atom and add temporal attention to $T$ $I_{\text{frames}}$. The operation is as follows:

$$I^{Te}_{\text{video}(x,y,c)} = ME^{Te}_{(\lfloor x / (X / \sqrt{T}) \rfloor, \lfloor y / (Y / \sqrt{T}) \rfloor, c)} \tag{21}$$

In this way, the one temporal position embedding is added into $I_{\text{video}}$ grid.

Then we simply put the two together to get our basic formula:

$$I^{Sp\text{-}Te}_{\text{video}(x,y,c)} = I^{Sp}_{\text{video}(x,y,c)} + I^{Te}_{\text{video}(x,y,c)} \tag{22}$$

---

**Algorithm 1:** Compute Spatial-Temporal Position Embedding

---

**input** : $I_{\text{video}}(X, Y, C)$

**output:** $I_{\text{video}}^{Sp\text{-}Te}(X, Y, C)$

1 **for** $(x, y, c) \in (0 : X, 0 : Y, 0 : C)$ **do**

2     **if** $c < \lfloor \frac{C}{2} \rfloor$ **then**

3        $ME_{(x,y,c)} = sin(\frac{(x \bmod (X / \sqrt{T}))}{\Theta^{\frac{c}{C}}}) + cos(\frac{(y \bmod (Y / \sqrt{T}))}{\Theta^{\frac{c}{C}}})$

4     **else**

5        $ME_{(x,y,c)} = sin(\frac{((x \bmod (X / \sqrt{T})) \times X) + y}{\Theta^{\frac{c - \lfloor \frac{C}{2} \rfloor}{C}}}) + cos(\frac{((y \bmod (Y / \sqrt{T})) \times X) + y}{\Theta^{\frac{c - \lfloor \frac{C}{2} \rfloor}{C}}})$

6 $I_{\text{video}}^{Sp\text{-}Te} = I_{\text{video}} + ME$

---

However, when we observe this formula, we can find that both spatial and temporal position embedding depend on the sine and cosine values of the positions $(x, y)$. For some combinations of $(x, y)$, the following situations may occur:

$$x \bmod (X / \sqrt{T}) = \left\lfloor x / (X / \sqrt{T}) \right\rfloor \ \& \ y / (Y / \sqrt{T}) = \left\lfloor y \bmod (Y / \sqrt{T}) \right\rfloor \qquad (23)$$

We assume that $X / \sqrt{T} = k$ is a constant in the same settings in image diffusion, and $x$ is variable in position embedding, then the equation $x \bmod k = \left\lfloor x / (X / \sqrt{T}) \right\rfloor$ holds true when $x$ is an integer multiple of $k + 1$, where $k > 1$. Specifically, $x$ can be expressed as $x = n(k + 1)$, where $n$ is any non-negative integer. And also $y$.

Therefore, there are some combinations of $(x, y)$ that satisfy:

$$\frac{x \bmod (X / \sqrt{T})}{\Theta^{\frac{c}{C}}} = \frac{\left\lfloor x / (X / \sqrt{T}) \right\rfloor}{\Theta^{\frac{c}{C}}} \ \& \ \frac{y \bmod (Y / \sqrt{T})}{\Theta^{\frac{c}{C}}} = \frac{\left\lfloor y / (Y / \sqrt{T}) \right\rfloor}{\Theta^{\frac{c}{C}}} \qquad (24)$$

And considering the periodicity of sine and cosine functions: $sin(\theta + 2k\pi) = sin(\theta)$ and $cos(\theta + 2k\pi) = cos(\theta)$. Even if $x \bmod (X / \sqrt{T})$ and $y / (Y / \sqrt{T})$ are not exactly the same as $\left\lfloor x / (X / \sqrt{T}) \right\rfloor$ and $\left\lfloor y / (Y / \sqrt{T}) \right\rfloor$, they may produce repeated embedding as long as they reach the same angle within one period.

To avoid possible duplication when spatial and temporal position embeddings are superimposed at the same position, we treat spatial and temporal as two independent parts and add them to different channels:

$$I_{\text{video}(x,y,c)}^{Sp\text{-}Te} = \chi(c < \lfloor C/2 \rfloor) \cdot I_{\text{video}(x,y,c)}^{Sp} + \chi(c \geq \lfloor C/2 \rfloor) \cdot I_{\text{video}(x,y,c - \lfloor C/2 \rfloor)}^{Te} \qquad (25)$$

Where $\chi(A)$ represents a Boolean condition function. When $A$ is true, $\chi(A)$ equals 1; otherwise, $\chi(A)$ is 0. Considering the periodicity of trigonometric functions, if $\Theta \gg X$ and $\Theta \gg Y$, all mappings occur within a single period, preventing any repetition.

In summary, we use Equation 18 and 19 to generate spatial and temporal position embeddings, and then combine them using Equation 25. This results in our spatial-temporal position embedding method. Additionally, the pseudo-code of our spatial-temporal position embedding is shown in Algorithm 1.

# D COMPARED BASELINES

To demonstrate the effectiveness of our model, we selected some baseline models from many recent text-to-video generation models. These baselines include all video generation methods, including transformer or diffusion. These baselines are as follows:

CogVideo (Hong et al., 2022), MagicVideo (Zhou et al., 2022), VideoComposer (Wang et al., 2024), VideoFactory (Wang et al., 2023c), SimDA (Xing et al., 2024), Show-1 (Zhang et al.,

2023), VideoFusion (Luo et al., 2023), PYoCo (Ge et al., 2023), Video LDM (Blattmann et al., 2023), LVDM (He et al., 2022), VideoCrafter (Chen et al., 2023), VideoCrafter2 (Chen et al., 2024), ModelScope (Wang et al., 2023b), StreamingT2V (Henschel et al., 2024), Gen-l-video (Wang et al., 2023a), OpenSORA (Jiang et al., 2024), ViD-GPT (Gao et al., 2024)

## E   IMPLEMENTATION DETAILS

We performed our experiments on NVIDIA RTX 3090 or NVIDIA A6000 GPUs using Python 3.10.13, PyTorch 2.2.1, CUDA 12.4, CuDNN 8.9.2. We use 4 to 8 GPUs for each training and evaluation.

### E.1   SAMPLING

We tested the generated videos with an FPS of 30, using 50 inference steps for diffusion. We employed classifier-free guidance with a guidance strength of 7.0. The test datasets included all the MSR-VTT, UCF-101, and VC datasets. A video was generated for each prompt.

### E.2   TRAINING

To fully utilize the high-performance tensor cores available in NVIDIA Ampere GPUs, we use mixed-precision training (precision=16) in all our training runs. Specifically, we store all trainable parameters as 32-bit floating point (FP32) but temporarily cast them to 16-bit floating point (FP16) before evaluating the model. We store and process all activation tensors as FP16, except for the embedding network and the associated per-block linear layers, where we opt for FP32 due to their low computational cost.

During the model training process, we employ the DeepSpeed strategy (stage=2) and enable optimizer offloading to the CPU (offload_optimizer=True) to reduce memory usage effectively. This strategy allows us to train larger models on limited hardware resources. Additionally, we use the CPUAdam optimizer, provided by DeepSpeed, which performs optimization calculations on the CPU, further reducing the computational burden on the GPU. Specifically, we configure the optimizer with a base learning rate, betas set to (0.9, 0.9), and a weight decay of 0.03.

Furthermore, we utilize the DeepSpeed internal checkpointing feature to store partial gradients and other intermediate states during training. This helps manage memory efficiently and ensures smooth training progress.

### E.3   FVD

We calculate FVD (Unterthiner et al., 2018) using I3D (Carreira & Zisserman, 2017) pretrained video feature encoder. I3D is pertrained on the Kinetics Human Action Video dataset (Kay et al., 2017). FVD calculates the distribution difference between two data sets, with lower values indicating a smaller difference between the two distributions. In this context, it means that a lower FVD value signifies that the distribution of generated videos is more consistent with the distribution of videos in the test data set. However, within the same model, the size of the FVD indicator is significantly affected by the number of videos. As shown in Figure 17, for most models, when the number of videos exceeds 10k, the FVD value stabilizes and fluctuates by only about 1%. Therefore, we

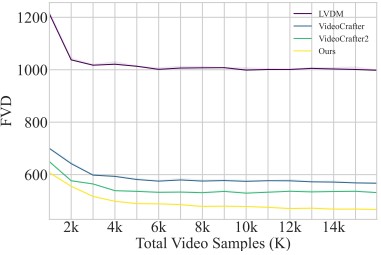

Figure 17: The different video samples calculated by FVD for different baselines on MSR-VTT.

select at least 10k videos for testing to ensure reliable and consistent FVD measurements. Additionally, to eliminate the differences caused by random seeds, we followed the experiment protocol by conducting tests with three different seeds and taking the minimum value as the experimental result. The remaining results are shown in the shaded areas in the figure.

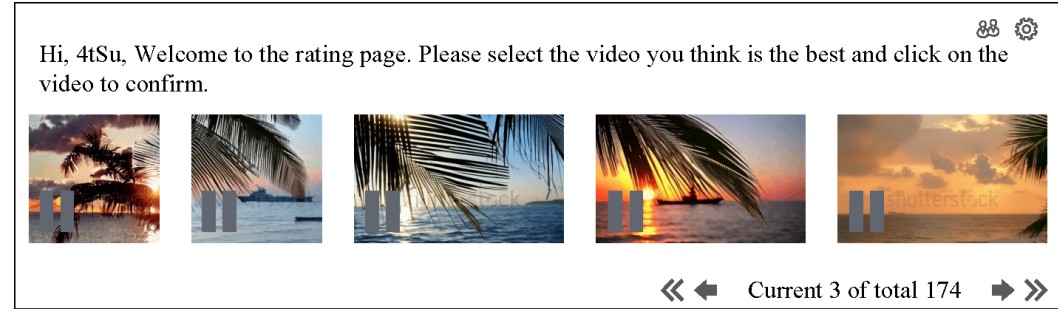

Figure 18: The sample of the user study website.

In MSR-VTT test, we use about 50k generated 16 frames of video clips and the whole MSR-VTT test dataset with randomly selected 16 frames of each video, without any augmentation such as horizontal flips. In UCF test, we use about 11k generated 16 frames of video clips and the whole UCF test dataset with randomly selected 16 frames of each video. Specifically, we use the label of each video for text input. The two video groups are calculated by:

$$\text{FVD} = \|\mu_r - \mu_g\|^2 + \text{Tr}\left(\Sigma_r + \Sigma_g - 2\left(\Sigma_r \Sigma_g\right)^{1/2}\right) \tag{26}$$

where $r$ and $g$ are multivariate Gaussians, which stand for the feature of test videos.

### E.4    TRAINING DATA

We preprocess the ImageNet, JDB, and WebVid datasets to ensure consistent training across all datasets. Since the training images and videos vary in resolution and are mostly non-square, we standardize the aspect ratio and resolution through the following steps:

1. The shorter edge of each image is resized to the target training resolution using bicubic interpolation.

2. A center crop is applied to ensure a square aspect ratio.

No additional data augmentations, such as horizontal flips, are used during training.

Additionally, for video datasets, we extract the required frames based on the FPS. For WebVid, we assume each video has a default frame rate of 60 FPS.

### E.5    MODEL PERFORMERS ESTIMATION

We use "TIME-python" to estimate the runtime of each block. To ensure accuracy, we constructed the same model in an identical inference environment, performed 1,000 inferences, and recorded the total time. To quickly obtain the inference time under different settings, we tested the same module by inputting a tensor that matches the specific shape required for each setting.

## F    USER STUDY

**Data Collection Method.** We asked volunteers to rate the results using an online website. The sample of website is shown on the figure  18. Volunteers were shown 5 videos generated by the same prompt and asked to choose the best one.

**Raw Collected Data.** We provided some raw data of the User Study as shown in the table  4. Most volunteers thought our video was the best. However, some volunteers such as wzmG, yCKy, and 2Wn6 thought VideoCrafter2 was better. However, the other three video generation models LVDM, VideoCrafter and ModelScope all scored low. However, for people related to artificial intelligence, they gave videocrafter2 a slightly higher score than ETC, but people working in the art industry gave ETC a high score. This shows that the generation quality of ETC may not be as good as VideoCrafter2, but the artistry may be stronger than VideoCrafter2.

| Profession | Volunteer Name | Volunteer Specific Occupation | LVDM | VideoCrafter | Model VideoCrafter2 | ModelScope | ETC |
|---|---|---|---|---|---|---|---|
| AI | cGNG | Multimodal Retrieval | 14.50% | 4.40% | 32.80% | 14.90% | **33.40%** |
| | wzmG | 3D Generation | 8.80% | 3.90% | **38.50%** | 13.40% | 35.40% |
| | WFpZ | Video Generation | 14.40% | 5.70% | **32.80%** | 12.70% | 24.40% |
| Art | cVfZ | Photography | 6.50% | 12.40% | 33.70% | 5.00% | **42.40%** |
| | iKtk | Video Edit | 6.90% | 11.50% | 28.50% | 16.40% | **36.70%** |
| | n12H | Painting | 0.70% | 17.20% | 10.30% | 14.40% | **57.40%** |
| | 99nF | Violin Making | 6.70% | 21.30% | 16.90% | 17.80% | **37.30%** |
| | yCky | Band Guitarist | 0.50% | 8.40% | **37.00%** | 20.40% | 33.70% |
| | 8TLu | Freelance Writer | 0.80% | 9.40% | 39.80% | 8.30% | **41.70%** |
| | 2Wn6 | Garden Design | 3.80% | 11.40% | **40.20%** | 12.10% | 32.50% |

Table 4: The scores of volunteers of different professions on different video models. The names of the volunteers have been replaced by a 4-digit random combination of uppercase and lowercase letters and numbers.

**Volunteer Background Information.** As for the participants in the user study, all of them are engaged in artificial intelligence multimodal research, generative fields, or traditional art fields. The art fields include video editing, graphic and image art creation, music composition, performance, photography, and other areas, with all participants having at least a bachelor's degree. About 40% of them have conducted research in the above interdisciplinary fields, around 60% have worked in these fields for more than two years, and approximately 30% have been involved for over five years. All participants signed consent forms for the user study. The consent form sample is shown in the figure 19.

## G  VC DATASET

The VC (Video Caption) dataset, which we developed, contains 500 prompts generated by GPT-4   OpenAI (2023). This dataset is designed to evaluate video generation performance using real-world open-domain prompts.

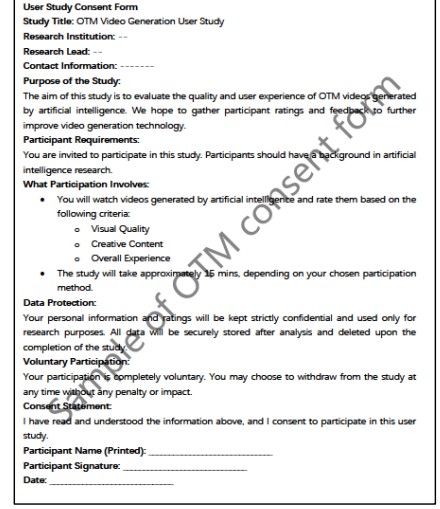

Figure 19: The sample of the consent form for the user study.

**Dataset Summary.** We first defined different open-domain scenarios (such as outer space, inside a cell, etc.) as well as various styles (anime style, realistic style, Van Gogh style, etc.). Then, we used ChatGPT to create detailed sentences suitable for video generation. During the generation process, GPT was asked to reference and summarize existing datasets, such as JDB. The average sentence length in this dataset is 40.3 words.

**Some Example of VC Dataset.** We randomly selected some sentences from the vc dataset, covering Space Exploration, urban scenery and other aspects. These sentences are shown in the table 5.

## H  AUTO ENCODER

To determine whether the autoencoder affects the stitched images, we conducted a study on a basic variational autoencoder trained with KL divergence. This module uses a combination of convolution and attention mechanisms to encode images into latents and decode latent representations back into images. We trained the module on the WebVid dataset for 54k iterations with a batch size of 2 with 2 loss functions:

| Stage | $\mathcal{L}_{\text{recon}}$ | $\mathcal{L}_{\text{KL}}$ | SSIM ↑ |
|---|---|---|---|
| Before Train | | | 0.897 |
| After Train | ✓ | | **0.921** |
| After Train | ✓ | ✓ | 0.914 |

Table 6: The SSIM before and after training.

$$\mathcal{L}_{\text{recon}} = \mathbb{E}_{q(z|x)}\left[\log p(x|z)\right] \quad (27)$$

| Category | Example Sentences |
|---|---|
| Space Exploration | - A spaceship travels through the star-filled galaxy, navigating the vastness of space.
- An astronaut floats in zero gravity, repairing equipment outside the International Space Station.
- A small asteroid spins rapidly in space, with Earth hanging far in the background.
- A rocket launches into space, leaving a trail of smoke as it breaks through the atmosphere. |
| Urban Landscapes | - The lights of skyscrapers flicker at night, with traffic flowing endlessly on city streets.
- A pedestrian walks across a bustling city square, with a towering TV tower in the distance.
- The setting sun casts a warm glow over old streets, as street lamps begin to light up, bringing calm and beauty to the city.
- A busy subway station with commuters rushing through, as trains arrive and depart in quick succession. |
| Natural Scenery | - A magnificent waterfall cascades down from the mountains, spraying mist as a rainbow forms in the water.
- A green meadow sways in the breeze, with flowers dancing under the sunlight and distant mountains bathed in warmth.
- Waves crash against rocky shores, the white foam sparkling under the setting sun.
- A forest blanketed in morning fog, birds chirping as the first light filters through the trees. |
| Fantasy Worlds | - A giant dragon soars through the sky, flames bursting from its mouth, with a forest stretching below.
- A wizard stands at the edge of a cliff, waving a glowing wand as the sky lights up with magical energy.
- An elven village hidden among giant trees, with soft light filtering through the leaves.
- A castle made of crystal floats in the sky, shimmering under the light of twin suns. |
| Underwater World | - Colorful schools of fish swim among coral reefs, the water crystal clear and calm.
- A giant blue whale moves slowly through the deep ocean, surrounded by drifting plankton.
- A diver explores a deep underwater cave, surrounded by glowing sea creatures.
- A pod of dolphins leap through the water, playing in the waves under a bright blue sky. |
| Futuristic Technology | - Autonomous flying cars navigate high-altitude routes, with a neon-lit city flashing in the distance.
- Robots efficiently work in a factory, with robotic arms rapidly assembling electronic devices.
- Humans interact with AI assistants in a virtual reality environment, creating precise 3D designs.
- A drone patrols a high-tech city, scanning for anomalies with its advanced sensors. |
| Microscopic World | - Under a microscope, a cell splits into two, the nucleus slowly dividing.
- A virus invades human cells, quickly replicating and spreading to neighboring healthy cells.
- Tiny organisms swim in a drop of water, propelled by their microscopic cilia.
- White blood cells move through the bloodstream, hunting down and attacking harmful bacteria. |
| Historical Scenes | - Roman gladiators fight in an ancient arena, as the crowd cheers loudly.
- A medieval knight procession passes through a castle gate, preparing for a grand celebration.
- Smokestacks of factories puff black smoke during the Industrial Revolution, with workers busy around machinery.
- Ancient Chinese scholars debate philosophy under a large pavilion by a riverbank. |
| Sci-Fi Adventures | - A space exploration team walks on the surface of an alien planet, discovering the ruins of an ancient alien civilization.
- A high-speed spaceship zips through the stars, encountering hostile alien forces mid-flight.
- An intelligent robot explores an unknown planet, collecting strange alien minerals.
- A group of space travelers ventures through a wormhole, arriving in a distant galaxy filled with unknown planets. |
| Surrealism | - A giant human hand rises from the ground, grasping the sun in the sky.
- Floating islands drift among clouds, with trees and houses suspended in mid-air.
- A school of fish swims through the sky, passing through colorful, rainbow-colored clouds while the ground below is a desert.
- An infinite staircase spirals up into the clouds, with people walking both up and down without ever reaching the end. |

Table 5: Some examples of vc dataset.

$$\mathcal{L}_{\text{KL}} = D_{KL}(q(z|x) \parallel p(z)) = \frac{1}{2} \sum \left( \sigma^2 + \mu^2 - \log(\sigma^2) - 1 \right) \tag{28}$$

The visual reconstruction before and after training is shown in the figure 20. As observed, even before training, the VAE, relying solely on convolutions, could already encode the stitched images and map tokens from different positions to the corresponding image areas during decoding. After training, there was no significant improvement in the reconstructed images. Quantitative results, as shown in the table 6, further support this finding. The SSIM value only increased by 0.017 after training with $\mathcal{L}_{\text{recon}}$ and $\mathcal{L}_{\text{KL}}$. If the KL divergence is not regularized, it may improve SSIM to 0.921 but will make subsequent diffusion training more difficult, which is why we chose not to train the VAE.

# I    MORE VISUAL RESULTS

In order to better demonstrate the visual effects, we show more high-resolution upscaling effects in figure 21, figure 22, and figure 23. These effects show that we have shown good quality in different styles, environments and other settings.

# J    NEGATIVE SOCIETAL IMPACTS

This work builds on Stable Diffusion v2 for its basic and improved models, with a few additional experiments using other versions of Stable Diffusion. As the quality of large visual generation models improves, they have increasingly led to negative societal impacts, such as making it easier and cheaper to produce fake news (Mishkin et al., 2022). Furthermore, the significant energy

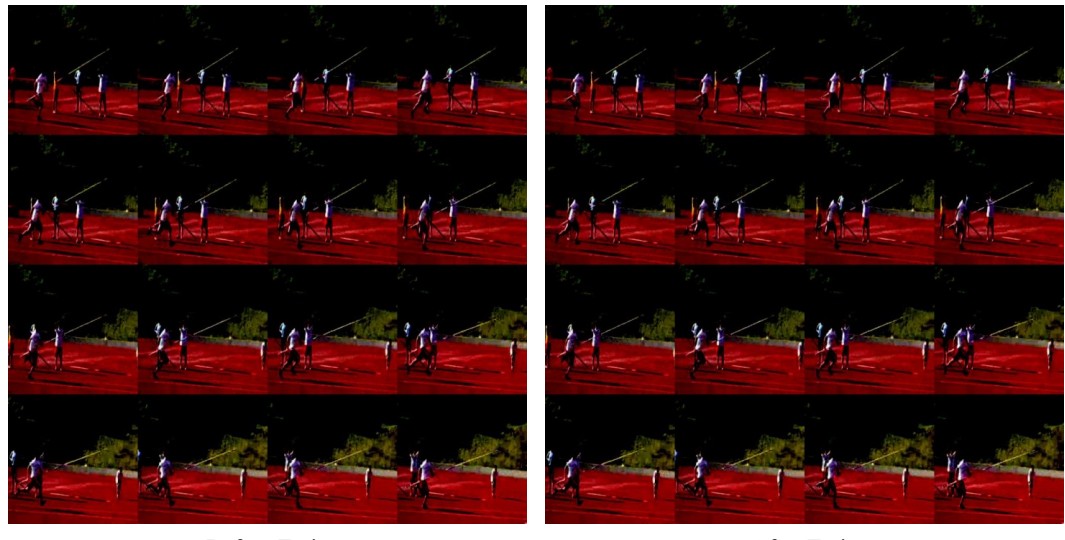

Before Train        After Train

Figure 20: Visualization of a video in MSR-VTT dataset compared with before and after VAE training.

consumption required for training and operating these large models can increase carbon emissions, potentially contributing to environmental issues.

Lisbon sunset bridge ship portugal motion timelaspe square city centre hyperlapse 25th of april bridge sunset

Happy little boy child running with toy airplane in wheat field at sunset. kid playing with airplane in summer nature outdoors. little pilot dreaming of flight, vacation with happy family, big dream.

Close up grill roast bbq chicken leg on the flaming grill , 4k

Yellow flowers are swaying by the wind in evening. common name indian laburnum or golden shower or pudding-pipe tree, scientific name cassia fistula linn

Girl blowing on a dandelion flower

Jordan, petra, december 5, 2016_ people near al khazneh or the treasury at ancient petra, originally known to nabataeans as raqmu - historical and archaeological city in hashemite kingdom of jordan

Spoon-billed sandpiper (calidris pygmaea), and other birds foraging side-to-side movement of the bill forward with its head down

Lisbon sunset bridge ship portugal motion timelaspe square city centre hyperlapse 25th of april bridge sunset

Professional male doctor having discussion with team on tablet

Atlanta, georgia capital, skyline, stars timelapse

Figure 21: More visual results for ETC.

Clean and renewable wind power farm in motion, turbine tower rotating, windmill field, green energy production, sustainable alternative electricity, no pollution environment

Vilnius, lithuania - june 19_ beautiful colourful modern laser event on river shore in city park on june 19, 2015 in vilnius, lithuania. 4k uhd video clip. zoom out

Snowdrop- spring white flower with bright shiny sun

Young woman in headphones listens to music. portrait closeup

Toddler girl in cute halloween dress looking for perfect pumpkin at the pumpkin patch

Slow motion of father and son running along the beach at sundown on a beautiful summer's evening. the father is holding his son's hand and they are ambling near ocean. slow motion. shot on red camera

A farmer uses a tablet to work in the field. a male agronomist analyzes the soil

Little cute girl making inhalation. girl studying the globe in her room. using a nebulizer and inhaler for treatment. indoor epidemic, self isolation, home quarantine, pandemic, coronavirus

Time-lapse of blooming apple tree branch 3b1 in png+ format with alpha transparency channel isolated on black background

Yellow lined sweetlips (plectorhinchus diagrammus) underneath coral reef at raja ampat

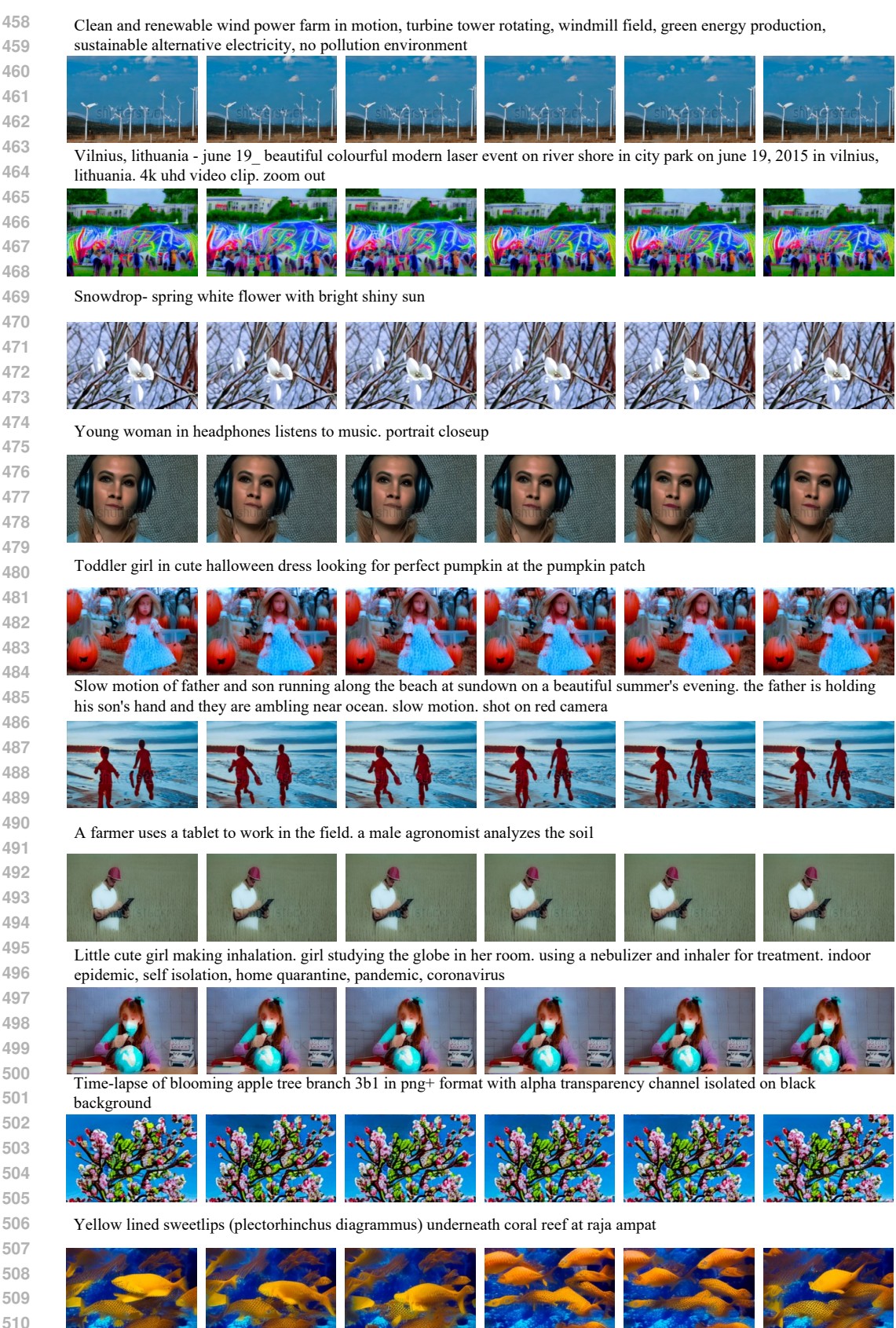

Figure 22: More visual results for ETC.

Butterflies and bees in the echinacea garden at sunny day. homeopathic herbs

Pink and white cosmos flower swaying in the garden on a clear sunny day. natural background. close up and colorful

Timelapse. ferris wheel on the background of a fast moving sky

Aerial view of seashore with beach, lagoons and coral reefs. philippines, luzon. ocean coastline with turquoise water. tropical landscape in asia

Time lapse_ dramatic stormy sky over dragon bridge in da nang, vietnam

abrador retriever dog is sleeping face closeup

On a morning summer meadow horse is grazed on leash. agricultural field on edge of a private farm

Squatter pigeon adult lone foraging

Holstein cow herd grazing, pasture, mountain farm, unpolluted grass, food

Beautiful girls in bright dresses throwing confetti at a party in a studio on a white background

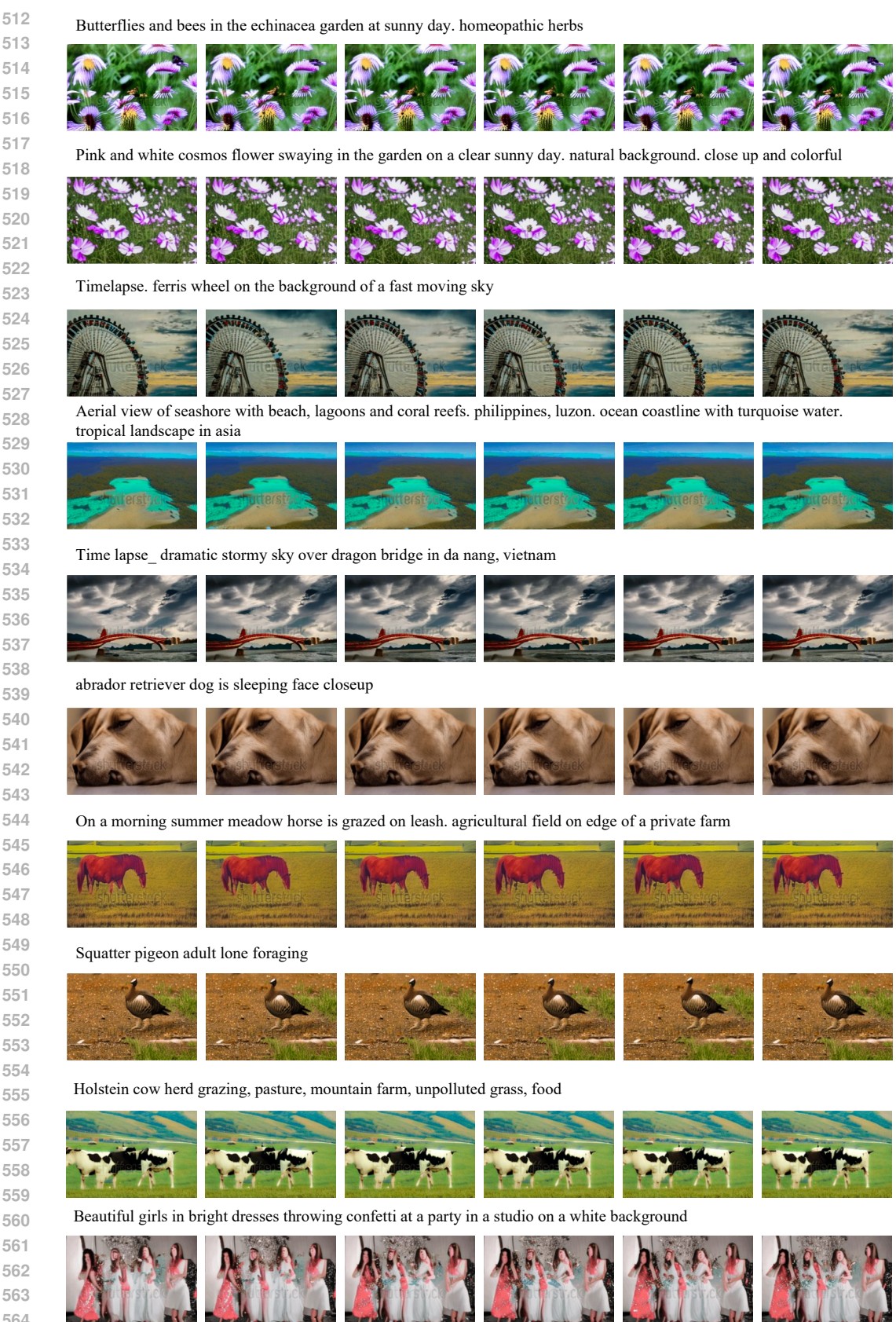

Figure 23: More visual results for ETC.

