# OpenReview forum: "ETC: Towards Training-Efficient Video Synthesis with Exploiting Temporal Capabilities of Spatial Attention"
_ICLR.cc/2025/Conference — ICLR 2025 Conference Withdrawn Submission_

### Official Review · Reviewer_KJcQ · 2024-10-28

**Soundness:** 2
**Presentation:** 2
**Contribution:** 2
**Rating:** 5
**Confidence:** 5

**Summary:**

This paper presents a training-efficient approach to train text-to-video (T2V) models. It explores how to transfer text-to-image (T2I) models to the T2V task without introducing a temporal model. Additionally, it proposes a data-efficient hybrid training method that allows the model to achieve favorable FVD metrics with relatively low training costs.

**Strengths:**

- The motivation and writing of this paper are very clear, making it easy to follow.
- From a quantitative perspective, the paper achieves good metrics at a relatively low training cost.

**Weaknesses:**

- The novelty is somewhat limited, as the approach in this paper aligns closely with [1], which also uses a grid-based approach to convert videos into images. The method in [1] originates from [2], which restricts the novelty of this paper.
- Although the paper proposes the spatial-temporal mixed embedding method, in essence, it is equivalent to adding a positional embedding. I am curious about how it prevents disrupting the T2I model’s weights at the beginning—this is an important point.
- The FPS embedding design is also not novel; it was first introduced in MagicVideo. The mixed FPS ranges from 0 (pure image) to 120 (single-frame replication). This design lacks significant originality.
- What bothers me most is the qualitative results. Although the quantitative metrics are promising, the qualitative results fall behind recent state-of-the-art video generation models like DiT architectures, OpenSora, Opensora-Plan, CogvideoX, etc. The failure cases, in particular, perform poorly.
- The paper does not validate any scaling laws in terms of data or model scalability.
- The authors should analyze more thoroughly where the quantitative advantages come from. Given the generally unimpressive visual quality, I can only assign a borderline rejection score for now.

References: [1] Lee T, Kwon S, Kim T. Grid Diffusion Models for Text-to-Video Generation [C] // Proceedings of the IEEE/CVF Conference on Computer Vision and Pattern Recognition. 2024: 8734-8743.

[2] Fan Q, Panda R. Can an image classifier suffice for action recognition? [J] arXiv preprint arXiv:2106.14104, 2021.

**Questions:**

See Questions in Weakness.

---

### Official Review · Reviewer_mANp · 2024-10-30

**Soundness:** 1
**Presentation:** 2
**Contribution:** 1
**Rating:** 3
**Confidence:** 4

**Summary:**

The paper introduces ETC, a framework aimed at training-efficient text-to-video (T2V) synthesis by exploiting spatial attention for temporal modeling. The authors propose to eliminate temporal attention layers, typically used in T2V models, by using spatial attention from pre-trained text-to-image (T2I) models. The framework introduces techniques like temporal-to-spatial transfer and spatial-temporal mixed embedding to handle video frames within a spatial grid. Extensive experiments demonstrate superior performance in terms of quality and efficiency over several state-of-the-art (SOTA) methods.

**Strengths:**

● The paper presents a new perspective by leveraging spatial attention for temporal modeling. It is interesting as this approach not only simplifies the architecture but also reduces training costs, providing new insights for video generation tasks.

● If all results are true under a fair comparison, the performance improvement is significant.

**Weaknesses:**

● It lacks convincing explanation for superior performance. While the authors attempt to explain why spatial attention can replace temporal attention, the reasons behind the significantly better results remain unconvincing. It is unclear why the proposed approach would outperform existing models to such an extent, especially considering the limited training resources used (8 NVIDIA 3090 GPUs).

● The model’s performance raises concerns about its generalization to more complex datasets or scenarios, especially given the small-scale training. The absence of detailed discussions about potential limitations, such as the restricted ability to model large motions due to implicit spatial relation modeling, weakens the validity of the results.

● Lack of visual evaluation. While the quantitative results are compelling, there is no video evaluation provided to visually demonstrate the effectiveness of the ETC framework. Also, the code in the supplementary materials is too basic to allow a direct assessment of the model’s qualitative improvements.

● In the supplementary materials, the authors claimed they include comparisons with many baselines, while the main paper does not provide sufficient detail on all these baselines or whether the comparisons were fair. This raises questions about the reported results, given that other well-recognized SOTA models typically use more data and computational resources. It would be beneficial to clarify how the proposed model achieves consistently the best results under such limited training conditions (as shown in Table 1).

**Questions:**

See the Weaknesses.

---

### Official Review · Reviewer_24a3 · 2024-10-31

**Soundness:** 1
**Presentation:** 2
**Contribution:** 1
**Rating:** 3
**Confidence:** 5

**Summary:**

This work aims at improving the data efficiency in training T2V models via reusing spatial attention for temporal modeling. In particular, the authors propose to rearrange a sequence of frames into a "stitched" huge frame. The authors claim that they achieve better synthesis quality than existing alternatives yet using less data.

**Strengths:**

- Studying the data efficiency in learning T2V models deserves a pat.

**Weaknesses:**

- From the motivation (or say theoretical foundation) part, I believe there exist **technical flaws**.

  - Intuitively, removing the temporal module and reusing the spatial module to handle both spatial and temporal information will definitely affect the model capacity. From this perspective, the so-called "temporal capabilities" of spatial attention does not convince me.
  - I will explain my concern with a toy example. Let $A = (a) \in \mathbb R^{1 \times 1}, B = (b_{ij}) \in \mathbb R^{2 \times 2}$, and $X = (x_1, x_2) \in \mathbb R^{1 \times 2}$. Assuming that $A, B$ are invertible, as required by the authors, there does **not** exist $A' = (a')$ such that $AXB=A'X$ for any $X$. First, note that $AXB = (a(b_{11}x_1 + b_{21}x_2), a(b_{12}x_1 + b_{22}x_2))$, and $A'X = (a'x_1, a'x_2)$. Then if $b_{11} = b_{22} = 0$, $AXB = (ab_{21}x_2, ab_{12}x_1)$ cannot be equal to $A'X = (a'x_1, a'x_2)$ for any $x_1 \neq x_2$. This clearly contradicts the claim in Line 878, which means **the theoretical foundation of this work does not hold**.

- From the empirical part, the quality of videos generated by ETC are not as good as those generated by previous approaches. I believe the reason is just that the modeling capacity of spatial attention struggles to handle the temporal information.

   - The frames in the last row of Figure 5 and those in Figure 6a are blurry.
   - The motion in all presented videos seems to be really small (Figure 6b, Figure 21, Figure 22, Figure 23).
   - There are even no videos provided in the supplementary material, which is very strange for a submission working on video synthesis.
   - Given the above observations, I wonder why the FVD metric from ETC is so small compared to other competitors.

**Questions:**

Please refer to the two major concerns listed in **Weaknesses**.

---

### Official Review · Reviewer_h2gj · 2024-11-01

**Soundness:** 2
**Presentation:** 3
**Contribution:** 2
**Rating:** 5
**Confidence:** 5

**Summary:**

This paper demonstrates that the spatial attention in T2I has a strong capability of temporal modeling and can boost the efficiency of training. Furthermore, this paper also propose a training-efficient framework, called ETC.

**Strengths:**

1. This paper discusses how to generate high-quality videos using only a pre-trained text-to-image model, which is very interesting.
2. The structure of this paper is well-organized and easy to follow.
3. The experimental results show the effectiveness of the proposed method.

**Weaknesses:**

There are some questions.
1. In the area of text-to-video generation, GridDiff adopts a similar approach. What distinguishes this work from GridDiff?
2. In lines 836 and 837, the authors claim that the primary components in the attention mechanism are linear operations. However, there are also some non-linear layers present in the whole network. If we take these non-linear layers into account, do equations (9) through (13) still hold?
3. In lines 191 to 192, the authors claim that single spatial attention has a larger receptive field than spatial and temporal attention combined. However, I think it is not appropriate to consider spatial and temporal attention in isolation from the rest of the network. If spatial and temporal attention are treated as a unified block for video modeling, would their receptive field still be considered smaller?
4. From Section 4, it appears that all video frames should be arranged into a single grid image. However, in Figure 3(a), there seem to be empty spaces. Why is this?
5. In the Spatial-Temporal Mixed Embedding section, the authors use absolute positional encoding. If the goal is to generate videos of varying resolutions and different video lengths, would it be necessary to include videos with diverse resolutions during the training phase?
6. For a more comprehensive quantitative evaluation of video generation, I recommend that the authors use a broader set of metrics, such as Vbench. Additionally, I suggest that the authors provide a video demo, allowing reviewers to more intuitively assess the quality of the generated videos.

**Questions:**

Please see above. If the author solves my problems, I will consider raising the score. Thanks.

---

### Official Review · Reviewer_sEp1 · 2024-11-09

**Soundness:** 2
**Presentation:** 3
**Contribution:** 2
**Rating:** 5
**Confidence:** 5

**Summary:**

The paper introduces ETC, a novel text-to-video synthesis model focused on training efficiency by exploiting spatial attention for temporal modeling. Unlike existing models that add temporal attention layers, ETC leverages only spatial attention with a temporal-to-spatial transfer strategy and spatial-temporal mixed embedding. This design reduces data dependency, allowing high-quality, efficient video generation using significantly smaller datasets.

**Strengths:**

- Proposes a highly efficient framework that eliminates temporal attention, reducing computational cost, which is an interesting idea.
- Innovatively uses a temporal-to-spatial transfer strategy and spatial-temporal embedding to enable video generation without sacrificing temporal consistency.
- Demonstrates superior performance with fewer training samples, achieving quality comparable to or better than current state-of-the-art methods.

**Weaknesses:**

- The authors use filtered high-quality video data to train their model, whereas the baseline methods do not incorporate this filtration step, potentially creating an uneven comparison. This difference in data quality could give the proposed model an advantage that does not solely stem from its architectural innovations.
- The paper claims that “We demonstrate that spatial attention modeling a linear mapping and alternating between spatial and temporal attention modeling another linear mapping, which does not model complex derivative or quadratic relationships.” However, this statement does not fully consider the inherent non-linearities of the model, nor does it account for the potential effects of stacking multiple spatial-temporal layers, which could enhance the model’s capacity to capture more complex relationships, including quadratic ones.
- Limited exploration of possible visual artifacts that may arise from removing explicit temporal modeling layers leaves open questions regarding the visual consistency and quality of generated videos. Additionally, relying primarily on FVD and CLIP scores limits the evaluation, as these metrics do not adequately capture human preference for smooth and realistic motion in videos. More human-centric evaluation metrics would improve the assessment of model performance.

**Questions:**

In Figure 3, why is it necessary to rearrange frames of videos into a single image?

---

### Note · Authors · 2024-11-14

I have read and agree with the venue's withdrawal policy on behalf of myself and my co-authors.